# Plant-Derived Natural Products in Cancer Research: Extraction, Mechanism of Action, and Drug Formulation

**DOI:** 10.3390/molecules25225319

**Published:** 2020-11-14

**Authors:** Wamidh H. Talib, Izzeddin Alsalahat, Safa Daoud, Reem Fawaz Abutayeh, Asma Ismail Mahmod

**Affiliations:** 1Department of Clinical Pharmacy and Therapeutics, Applied Science Private University, Amman 11931, Jordan; asmamahmod1212@gmail.com; 2Department of Pharmaceutical Chemistry and Pharmacognosy, Applied Science Private University, Amman 11931, Jordan; i_alsalahat@asu.edu.jo (I.A.); s_daoud@asu.edu.jo (S.D.); r_abutayeh@asu.edu.jo (R.F.A.)

**Keywords:** alternative anticancer therapies, natural products, plant extracts, curcumin, thymoquinon

## Abstract

Cancer is one of the main causes of death globally and considered as a major challenge for the public health system. The high toxicity and the lack of selectivity of conventional anticancer therapies make the search for alternative treatments a priority. In this review, we describe the main plant-derived natural products used as anticancer agents. Natural sources, extraction methods, anticancer mechanisms, clinical studies, and pharmaceutical formulation are discussed in this review. Studies covered by this review should provide a solid foundation for researchers and physicians to enhance basic and clinical research on developing alternative anticancer therapies.

## 1. Introduction

Cancer has been highlighted as one of the leading causes of death globally. Its incidence is in continuous rise, and an increase by 70% is expected over the next 20 years [1]. Conventional cancer therapies involve surgery, radiation, and chemotherapy. The use of chemotherapy is associated with cancer recurrence, emergence of resistance, and the development of severe side effects [2].

Plants have been considered for many years as an essential source of medicine to treat different ailments. One of the oldest records to use plant products in medicine come from clay tablets in cuneiform that were created by Sumerians in Mesopotamia (2600 BC). These tablets showed the use of more than 1000 plant-based products in medical treatment [3]. The use of plants to treat diseases was also popular among ancient Egyptians. Historical records revealed the use of more than 700 plant-derived products in medical treatments [4].

The limited efficiency and serious side effects associated with the use of conventional anticancer therapies encouraged scientists to focus on the discovery and development of new anticancer agents derived from natural products [5]. Secondary metabolites from plant sources like flavonoids, alkaloids, terpenoids, saponins, and others have been reported as important sources for potent anticancer agents [6,7,8,9]. The majority (more than 60%) of anticancer drugs that showed high efficiency in clinical use was obtained from plants, aquatic organisms, and microorganisms. The anticancer effect of these natural products is mediated by different mechanisms, including apopotosis induction, immune system modulation, and angiogenesis inhibition [10].

In this review, we summarize 14 anticancer agents derived from plants. A comprehensive discussion was provided to cover their natural sources, extraction methods, mechanisms of action as anticancer agents, their use in clinical trials, and pharmaceutical formulation.

## 2. Plant-Derived Natural Products as Anticancer Agents

### 2.1. Curcumin

Curcumin is one of three components of diferuloylmethane phenolic compounds known as curcuminoids. It is a major active constituent found in the dried rhizomes of *Curcuma longa* (family: Zingiberaceae), which is commonly known as turmeric [11,12,13,14,15,16]. The chemical structure was first identified by Lamp and Milobedeska in 1910 [15,17,18,19] (Figure 1). It has two aromatic *O*-methoxy phenolic groups, a β-dicarbonyl moiety and a seven-carbon linker containing two enone moieties; its IUPAC name is (1*E*,6*E*)-1,7-bis(4-hydroxy-3-methoxyphenyl)-1,6-heptadiene-3,5-dione) [15,16].

Curcumin was extracted and isolated for the first time by Vogel in the 19th century [15,17,18,19]. Application of various conventional methods for the extraction of curcuminoids from natural sources involves organic solvents extraction, steam distillation, hot and cold percolation, use of alkaline solution [13], and use of hydrotrope [20]. Moreover, several advanced methods have been also studied, such as supercritical fluid extraction, which has the advantage of being free from organic solvents, ultrasonic and microwave-assisted extraction, and enzyme-assisted extraction [18,21,22]. The Soxhlet extraction is considered as the traditional reference method, and when compared to the more advanced methods, curcumin extraction yield using Soxhlet method was considerably higher than those obtained from microwave-assisted, ultrasound-assisted, and enzyme-assisted extractions [22]. Post-extraction processes mainly include chromatographic techniques to separate the curcuminoids from other co-extracted volatile oils and oleoresins and to isolate curcumin from its correspondent curcuminoid compounds, namely demethoxycurcumin and bisdemethoxycurcumin [23,24]. Several organic solvents have been used to extract curcumin, yet ethanol remains the preferred solvent [21], and food-grade solvents, such as triacylglycerols, are being trialed and employed [25].

The various developed methods aim to decrease the amount of organic solvents used in the extraction methods and to decrease time required for the multi-step extraction and post-extraction procedures, including separation of curcumin from its analogs. Additionally, they aim to find a more selective extraction method, with high quality yield for food and therapeutic purposes, that proves to be a cost-effective method [13,20,22,23,24,26].

Curcumin has been acknowledged to exhibit several pharmacological properties, including anti-inflammatory, antioxidant, antibacterial, antiviral, anti-diabetic, and wound-healing ability and is widely researched for its potential anticancer and chemopreventive activity against various types of cancer [11,15,16,17,27]. It produces its anticancer effect through different mechanisms of action that include the inhibition of cancer cell growth, induction of cancer cell apoptosis, and suppression of cancer cell metastasis. These mechanisms have been studied in vitro and in vivo in a wide variety of cancers, including colorectal and breast cancer [17,28,29,30], where curcumin was involved in several signaling pathways, including inducing tumor-necrosis-factor-related apoptosis inducing ligand (TRAIL) apoptotic pathways via upregulating death receptor 5 (DR5) in HCT-116 and HT-29 colon cancer cells [31]. Additionally, curcumin initiated Fas-mediated apoptotic pathway in HT-29 colon cancer via caspase 8 activation [32], and it was found to upregulate Bax expression and suppress Bcl-2 through the phosphorylation at Ser15 and activation of p53 in HT-29 colon adenocarcinoma cell [33], in HCT-116 [17] and COLO-205 cells [34]. Curcumin has also been reported to inhibit NF-κB-luciferase HT-29 and in HCT-116 colon cancer cells and to inhibit Wnt/β-catenin pathway in vitro in HCT-116 colon cancer cells and in vivo in mice carrying APC gene mutation.

The anticancer effect of curcumin on osteosarcoma was mediated by inactivation of JAK/STAT signaling and inhibition the proliferation and migration of MG-63 cells [14]. Curcumin interferes with a number of cellular pathways (in vivo and in vitro) in prostate cancer, including mitogen-activated protein kinase (MAPK), epidermal growth factor receptor (EGFR), and nuclear factor κ (NFκB) [19]. Moreover, curcumin regulates p53 protein in vivo and in vitro in several breast cancer cell lines as reviewed by Talib et al. (2018) [15]. Curcumin modulates cellular pathways involved in cell proliferation of head and neck squamous cell carcinoma, most notably NF-κB and STAT3, which are found to be overexpressed in several head and neck carcinomas [19]. In vivo study using human glioma U-87 cells xenografted into athymic mice showed that curcumin is able to suppress glioma angiogenesis through inhibiting MMP-9 and downregulating endothelial cell markers. Curcumin was also able to induce G2/M cell cycle arrest by increasing protein kinase 1 (DAPK1) in U-251 malignant glioblastoma cells, which indicates that suppressing DAPK1 by curcumin does not only induce cell arrest but also inhibits STAT3 and NF-κB and activates caspase-3 [19].

### 2.2. Resveratrol

Resveratrol is a naturally occurring polyphenol that belongs to the stilbene class [35]. It is extracted from different types of plants and presented in 34 families involving 100 species [36]. High concentrations of resveratrol have been found in peanuts, soybeans, purple grapes, and pomegranates [37]. Although resveratrol has *cis* and *trans* configurations, the *trans* isomer is more stable with high bioactive effects [38]. Resveratrol is mainly extracted from roots, leaves, flowers, fruits, and seeds [36]. Different methods for extraction and separation of resveratrol have been reported. Organic solvent extraction is one of the conventional procedures used to extract resveratrol [39,40]. A novel enzyme-assisted ultrasonic method was applied to extract resveratrol from *Polygonum cuspidatum*. It produced a significantly high yield of 11.88 mg/g [41]. Moreover, *trans*-resveratrol extraction from peanut sprouts was optimized via using accelerated solvent extraction, and the response surface method [42]. A comparative study has shown that maceration method produced a high yield of resveratrol compared to ultrasound-assisted extraction and microwave-assisted extraction [43]. Resveratrol in red wine was determined using the online solid-phase extraction HPLC method improved by exerting a novel nanofibrous sorbent [44]. Another study has shown a significant enhancement of the total yield of resveratrol by applying thermal heating followed by enzymatic treatment (β-glucanase and pectinases) of grape peel extracts [45]. Several peanut oils with different brands from the local market were analyzed to determine *trans* resveratrol. The study revealed that using rapid magnetic solid-phase extraction based on alendronate sodium grafted mesoporous magnetic nanoparticles may effectively detect *trans*-resveratrol [46]. 

Resveratrol (3,5,4′-Trihydroxystilbene) is a stilbenoid and a phytoalexin produced by several plants in response to injury or any pathogen attack [47]. The basic structure of resveratrol is composed of two phenolic rings bonded together by a double styrene bond (Figure 2) [48]. Resveratrol has a low absorption rate due to low water solubility related to its chemical structure [49]. In the past, resveratrol has been used for stomachache, hepatitis, arthritis, urinary tract infections, and inflammatory and cardiovascular diseases [50].

Recently, several studies have focused on the anticancer properties of resveratrol and revealed its high ability to target multiple cancer hallmarks [51]. Resveratrol has displayed apoptotic and antiproliferative effects on human cervical carcinoma cells by inhibiting cell growth, activating caspase-3 and caspase-9, upregulating of Bcl-2 associated X protein, and inducing expression of p53 [52]. Moreover, resveratrol inhibited colon cancer cell proliferation, induced cell apoptosis, and G_1_ phase arrest via suppression of AKT/STAT3 signaling pathway [53]. Another study has shown that resveratrol improved apoptotic and oxidant effects of paclitaxel by activating TRPM2 channel in glioblastoma cells [54]. Additionally, resveratrol exhibited a cytotoxic effect against head and neck squamous cell carcinoma and reduced vascular endothelial growth factor (VEGF) expression [55]. Resveratrol inhibits metastasis in pancreatic cancer cells by affecting IL-1β, TNF-α, vimentin, N-cadherin, and CTA-2 expressions [56]. Zhao et al. reported that encapsulated resveratrol within peptide liposomes has improved the physicochemical properties and greatly reduced the toxicity of free resveratrol. It induced apoptosis in breast tumor by upregulating p53 and Bax expression, increasing Bcl-2 activity, and inducing caspase-3 activation [57].

Chatterjee et al. has found that resveratrol and pterostilbene are effective in remarkably shrinking a cervical cancer tumor model in vivo when injected directly into the tumor [58]. A combination of resveratrol and thymoquinone has been investigated in both models in vitro and in vivo, the results showed significant inhibition of cancer cells, promotion of apoptosis, and suppression of angiogenesis [59,60]. Another study revealed a synergistic effect between resveratrol and doxorubicin against breast cancer cells. Combination therapy inhibited tumor volume and increased life span in Ehrlich ascetic carcinoma cells bearing mice [61]. Furthermore, *trans*-resveratrol exhibited antitumor activity on human melanoma cells in a dose-dependent manner [62]. Cheng et al. reported that resveratrol induced cellular reactive oxygen species accumulation resulted in apoptosis activation and inhibit the proliferation of pancreatic cancer cells [63]. Recently, bovine serum albumin coated layered double hydroxide (LDH-BSA) was used to encapsulate resveratrol. The nanohybrid’s anticancer ability was investigated in human lung cancer cells (A549) and indicated higher activity comparing to bare resveratrol [64]. Moreover, resveratrol initiates the apoptosis and autophagic death of lung cancer cells by stimulating p53 signaling pathway [65].

### 2.3. EGCG (Epigallocatechin Gallate)

Epigallocatechin-3-gallate (EGCG) is a natural polyphenol that belongs to the flavonol class [66]. The main dietary sources of EGCG is green tea (*Camellia sinensis*, Theaceae) [67] and cocoa-based products [68]. Various extraction methods have been used to extract bioactive compounds from green tea, such as conventional solvent extraction, microwave-assisted extraction, ultrasonic-assisted extraction, supercritical carbon dioxide, Soxhlet extraction, high-pressure processing, and subcritical water extraction [69,70,71]. Modulation conditions of the ultrasound-assisted extraction method have optimized the extracted amount of EGCG from lipid extracted microalgae [72]. Moreover, subcritical water extraction of EGCG from green tea has been applied with adjusted extraction parameters resulted in a 4.66% yield of EGCG [73]. The extraction efficiency of epigallocatechin gallate was improved by using electrochemical methods. It has been found that using polymeric electrode PAN/PPY enriched with nanoparticles of TiO_2_ and rGO has saved time and increased efficiency to extract high-purity EGCG [74]. Ayyildiz et al. have shown that the ultrasound-assisted method was more efficient in extracting EGCG than the hot water method; however, it could be used for the production of green tea beverages [75]. Furthermore, using a green extracting agent like β-cyclodextrin improved the extraction yield of EGCG and ECG, compared to the water and ethanol solvent [76]. 

Epigallocatechin-3-gallate is the ester of epigallocatechin and gallic acid (Figure 3) [77]. Traditionally, green tea has been used in Chines and Indian medicine as a stimulant, diuretic, astringent, and to improve heart health [78]. The EGCG has various health benefits represented by reducing LDL cholesterol levels, inhibiting the abnormal formation of blood clots, suppressing tumor growth [79]. Among the different green tea catechin derivatives, EGCG is the most potent anti-inflammatory and anticancer agent [80].

Several studies have shown the EGCG properties as an anticancer agent. It has antiproliferative, antimetastasis, and pro-apoptosis activities [81]. EGCG inhibited the metastatic activity of human nasopharyngeal carcinoma cells by downregulation of protein expression of MMP-2 through modulation of the Src signaling pathway [82]. Moreover, combining EGCG with eugenol or amarogentin exhibited synergistic chemotherapeutic potential in the cervical cancer cell line. The antiproliferative effect was justified by their ability to downregulate cyclinD1 and upregulate of cell cycle inhibitors LIMD1, RBSP3, and p16 at G1/S phase of the cell cycle [83]. Naponelli et al. reported that EGCG induced endoplasmic reticulum stress affected gene expression, and interfere with intracellular proteostasis at different levels [84]. Furthermore, EGCG was able to sensitize cisplatin-resistant oral cancer CAR cell apoptosis and autophagy by activating AKT/STAT3 pathway and suppressing multidrug resistance 1 signaling [85]. Several in vivo studies have investigated the effect of consuming green tea on the reduction of incidence of malignant tumors, including colorectal, stomach, liver, and lung cancer [86,87]. Interestingly, ten different polyphenols have been tested to determine their chemopreventive activity, EGCG showed the most potent antiproliferative effects, and significantly stimulated cell cycle arrest in the G1 phase and cell apoptosis [88]. Chen et al. reported that EGCG nanoemulsion may inhibit lung cancer cells through matrix metalloproteinase (MMP)-2- and -9-independent mechanisms [89].

Sheng et al. demonstrated the effect of EGCG on doxorubicin-induced oral keratinocyte cytotoxicity and anticancer activity against oral cancer cells. It mitigated the cytotoxic effect of doxorubicin without weakening its anticancer efficacy [90]. Moreover, EGCG was able to suppress tumor growth of prostate cancer in TRAMP mice and decreased tumor-derived serum PSA [91]. A synergistic anticancer activity of curcumin and catechins was reported against human colon adenocarcinoma and human larynx carcinoma cell lines [92]. Recently, PLGA-encapsulated epigallocatechin gallate (EGCG-NPs) showed higher activity than free EGCG in inhibiting lung cancer tumors in PDX model by suppressing the expression of NF-κB regulated genes [93]. Additionally, epigallocatechin-3-gallate-loaded gold nanoparticles exhibited significant anticancer efficacy in Ehrlich ascites carcinoma-bearing mice [94].

### 2.4. Allicin

Allicin is a thioester of sulfenic acid or allyl thiosulfinate (Figure 4). It is found mainly in garlic (*Allium sativum*) and belongs to the Liliacerae family [95]. Allicin is chemically unstable and easily decomposed into oil-solubles such as diallyl sulfide, diallyl disulfide, and diallyl trisulfide, as well as water-solubles such as SAC and *S*-allyl mercaptocysteine [96]. Hence, the processing conditions have high impact on the composition of thiosulfinates compounds in garlic [97]. Shi et al. reported that spray-drying, freeze-drying, and oven-drying at a high temperature of fresh garlic resulted in a loss of activity and damaging of the alliinase which led to prevent allicin formation [98]. Allicin-rich extract has obtained from garlic by pressurized liquid extraction with a concentration of 332 μg of allicin per gram of sample. This method was more efficient compared to fresh garlic and garlic powder samples [99]. Moreover, using other extraction methods were reported such as supercritical CO_2_ extraction [100], supercritical fluid extraction [101], HPLC-MTT assay [102], and ultrasonic-assisted extraction [103]. Li et al. showed that applying salting-out extraction with optimized conditions, produced allicin with high purity (68.4%), compared to the purity of crude extract (31.8%) [104]. Recently, allicin was extracted with water then ultrasound-assisted binding with whey protein isolates to form conjugates. This process enhanced the stability, solubility, and emulsifying properties of allicin [105].

Alliin is a sulfoxide that represents 80% of the cysteine sulfoxides in garlic and is considered the allicin precursor molecule. The Allinase enzyme activated when garlic bulbs crushed or injured resulted in a conversion of alliin into allicin [95]. For ages, garlic has been used to cure many diseases, including hypertension, infections, and snake bites [106].

Allicin has been shown to possess different biological activities, such as anti-inflammatory, antimicrobial, and anticancer. Chen et al. reported that allicin significantly suppressed cell proliferation and invasion of cholangiocarcinoma cells. It induced apoptosis and prevented cell migration through upregulating of SHP-1 and inhibiting STAT3 activation. Moreover, it attenuated tumor growth in the nude mouse model of cholangiocarcinoma [107]. Furthermore, an in vivo study has been conducted to evaluate allicin effect on the radiosensitivity of colorectal cancer cells. The results showed that allicin enhances the sensitivity of X-ray radiotherapy in colorectal cancer via inhibition of NF-κB signaling pathway [108]. Besides, allicin exhibited antitumor activity against HCMV-infected glioma cells via inhibition of cytokine release, upregulation of p53 activity, and sensitivity improvement to radiotherapy [109]. It is also found that allicin could upregulate miR-486-3p and increase chemosensitivity to temozolomide in vitro and in vivo [110]. Schultz et al. demonstrated the activity of allicin in inhibiting ornithine decarboxylase, a rate-limiting enzyme in cell proliferation of neuroblastoma, and inducing cell apoptosis [111]. Moreover, allicin suppresses melanoma cell growth via increasing cyclin D1 and reducing MMP-9 mRNA expression [112]. It also inhibits human glioblastoma proliferation by stimulating S and G_2_/M phase cell cycle arrest, apoptosis, and autophagy [113]. Allicin showed efficacy in reducing growth and metastasis of gastric carcinoma through upregulation of miR-383-5p and downregulation of ERBB4 [114].

Allicin revealed a synergistic anticancer activity with 5-fluorouracil against lung and colorectal carcinoma cells [115], as well as sensitizing hepatocellular cancer cells to 5-fluorouracil [116]. It was reported that allicin can effectively hinder cell growth of U251 glioma cells [117] and reduces tumor burden in breast cancer cells [118,119].

### 2.5. Emodin

Emodin is most commonly extracted from the roots and rhizomes of, *Rheum palmatum* (Chinese rhubarb, family: Polygonaceae), although it is found in other plants from the same family, such as *Polygonum cuspidatum* (Asian knotweed) and *Polygonum multiflorum* (Chinese knotweed), and in plants from other families, namely *Aloe vera* (family: Asphodelaceae) and *Cassia obtusifolia* (Chinese senna, family: Fabaceae) [120,121,122]. It is also isolated from different fungal species, including *Aspergillus ochraceus* and *Aspergillus wentii* [123]. Emodin (1,3,8-trihydroxy-6-methyl-anthraquinone), is a natural anthraquinone derivative [124] (Figure 5), known to have various therapeutic activities, such as antibacterial, anti-inflammatory, antiviral, antitumor, immunosuppressive, and other pharmacological activities [125,126,127,128].

The methods for emodin extraction from herbs have included maceration extraction (ME), reflux extraction (RE), ultrasonic nebulization extraction (UNE) microwave-assisted extraction (MAE), stirring extraction (SE), supercritical carbon dioxide extraction and preparative liquid chromatography [123,129,130,131]. ME procedure is a very simple extraction method that could be used for the extraction of thermo-labile components. Nevertheless, this method is time-consuming with low extraction yield [132,133]. RE technique does not need as much time as ME, and it consumes smaller amounts of solvent. However, RE can only be used to extract thermo-stable chemicals [133,134]. Ultrasonication extraction UE is an extraction method that uses ultrasonic wave energy, where these waves produce cavitation in the solvent accelerating the dissolution and diffusion of the solute, as well as the heat transfer. UE could be applied to the extraction of thermo-labile compounds using small amounts of solvent with low energy consumption. This approach is commonly employed to extract polyphenols, ginsenosides, and other natural compounds. Moreover, it is a time-saving procedure and convenient operation that results in high extract yield [131,133,135].

UNE is a viable and alternate method for extraction from plant samples with proper constituents. UNE is different from UE because it uses aerosols carried by gas. This approach has many advantages over the other methods, because it usually gives the highest extract yield while still saving time [131]. Solid-phase extraction method might be employed to isolate emodin from red pigment mixture produced by the *A. ochraceus* [123].

According to Hsu and Chung’s review (2012), the molecular mechanisms of emodin comprise cell cycle arrest, apoptosis, and the promotion of the expression of hypoxia-inducible factor 1α, glutathione *S*-transferase *P*,*N*-acetyltransferase, and glutathione phase I and II detoxification enzymes while inhibiting angiogenesis, invasion, migration, chemical-induced carcinogen-DNA adduct formation, HER2/neu, CKII kinase, and p34cdc2 kinase in human cancer cells [136]. It has been reported to inhibit tumor-associated angiogenesis through the inhibition of ERK phosphorylation. It also enjoys antiproliferative and antimetastatic effects [137]. It downregulates the expression of survivin and β-catenin, inducing DNA damage and inhibiting the expression of DNA repair [136,138]. It also inhibits the activity of casein kinase II (CKII) by competing at ATP-binding sites [136,139]. According to some findings, it upregulates hypoxia inducible factor HIF-1 and intracellular superoxide dismutases and boosts the efficacy of cytotoxic drugs [140,141].

Emodin may sensitize tumor cells to radiation therapy and chemotherapy and inhibit the pathways that lead to treatment resistance. It was found to reverse gemcitabine resistance in vitro in pancreatic cancer cell lines by decreasing the expression of MDR-1 (*P*-gp), NF-κB, and Bcl-2 and increasing the expression levels of Bax, cytochrome-C, and caspase-9 and -3, and promoting cell apoptosis unstimulated and in gemcitabine-induced-resistance pancreatic cancer cell lines [142]. Furthermore, in vitro and in vivo findings concluded that emodin downregulated both XIAP and NF-κB and enhanced apoptosis in mice bearing human pancreatic cancer cells [143,144]. Chemosensitization was also observed in gallbladder cancer, where independent combination treatment of emodin with cisplatin, carboplatin, or oxaliplatin augmented chemosensitivity in vitro in SGC996 gallbladder cancer cells and in vivo in gallbladder tumor-bearing mice. Wang et al. (2010) credited these findings to the reduced glutathione level, and downregulation of multidrug resistance-related protein 1 (MRP1), and to the increased apoptosis caused by such combinations [145]. Additionally, enhanced chemosensitivity was observed in vitro in DU-145 cancer cell lines (multidrug resistant prostate carcinoma cell line) and in vivo in tumor-bearing mice when treated with a combination of emodin and cisplatin. The mechanism was shown to involve ROS-mediated suppression of multidrug resistance and hypoxia inducible factor-1 in over activated HIF-1 cells [146].

### 2.6. Thymoquinone (TQ)

Thymoquinone (TQ) is the main phytochemical bioactive constituent found in the volatile oil isolated from the *Nigella sativa* (black cumin, black seed), which has been used as a traditional medicine in many countries [59,147,148]. TQ has many pharmacological activities, including antioxidant, anti-inflammatory, immunomodulatory, antihistaminic, and antimicrobial, as well as with very promising antitumor activity [148,149,150,151,152] (Figure 6).

TQ can be obtained by different extraction methods such as hydrodistillation (HD), using Clevenger-type apparatus, dry steam distillation (SD), steam distillation of crude oils obtained by solvent extraction (SE-SD), and supercritical fluid extraction (SFE-SD). In both HD and SD, the extraction process is completed when pale yellow oil is formed [153]. SE is typically carried out with a Soxhlet apparatus, using *n*-hexane as a solvent. After 120 h of extraction, the residue is subjected to steam distillation and an additional extraction step to be followed with rotary evaporation, which produces a brownish yellow volatile oil [153]. SFE is a flexible extraction method due to the possibility of continuous modulation of the solvent. Different solvents can be used in this method, such as hexane, pentane, butane, nitrous oxide, sulphur hexafluoride, and fluorinated hydrocarbons. However, the most common SFE solvent used is carbon dioxide (CO_2_) since it is cheap, available, and safe. The extraction must be at low pressures and the temperature must be close to room temperature. Still, this method has one drawback which is the higher investment costs compared to traditional atmospheric pressure extraction techniques [129,153].

TQ potential anticancer activities is mediated by several mechanisms that alter the regulation of cell cycle, growth factor, protein kinase enzyme, tumor-suppressor gene, apoptosis, survival signals, transcription factors, and phase I and II enzymes [148]. Altering of cell cycle progression is an important step in the inhibition of cancer development and progression. TQ conjugated with fatty acid has potential activity on cell proliferation, apoptosis, and signaling pathways [148]. Conjugation is done to increase TQ’s capacity to penetrate cell membranes. Several conjugated forms were studied in HCT116 and HCT116 p53^−/−^ colon cancer and HepG2 hepatoma cells in vitro. Treatment with TQ-4-α-linolenoylhydrazone or TQ-4-palmitoylhydrazone was effective in p53-competent HCT116 cells, mediated by an upregulation of p21cip1/waf1 and a downregulation of cyclin E, and associated with an S/G2 arrest of the cell cycle. HCT116 p53^−/−^ and HepG2 cells showed only a minor response to TQ-4-α-linolenoylhydrazone [154]. TQ induced the G0/G1 cell cycle arrest, increased the expression of p16, decreased the expression of cyclin D1 protein in DMBA-initiated TPA-promoted skin tumors in mice, inactivated CHEK1, and contributed to apoptosis in colorectal cancer cells [155,156]. Moreover, TQ causes cell arrest at different stages according to concentration used (25 and 50 μM) in vivo in human mammary breast cancer epithelial cells line, MCF-7 [157]. TQ reduced the elevated levels of serum TNF-α, IL-6, and iNOS enzyme production and enhanced histopathological results in Wistar rats with methotrexate-induced injury to hepatorenal system [158]. Additionally, TQ has a role in reducing the NO levels by downregulation of the expression of iNos, reducing Cox-2 expression and consequently in generating PGE2 and reducing PDA cells synthesis of Cox-2 and MCP1 [159,160].

TQ has an effective role in the reduction of endothelial cell migration, tube formation and suppression of tumor angiogenesis. TQ noticeably reduced the phosphorylation of EGFR at tyrosine-1173 residues and JAK2 in vitro in HCT 116 human colon cancer cells [161]. TQ causes G2/M cell cycle arrest and stirred apoptosis, and it significantly lowers the nuclear expression of NF-κB. Moreover, TQ has a role in the elevation of PPAR-γ activity and downregulation of the gene’s expression for Bcl-2, Bcl-xL, and survivin [162]. Furthermore, it has an antiproliferative effect, especially, when combining it with doxorubicin and 5-fluorouracil which resulted in increased cytotoxicity in breast cancer xenograft mouse model [162]. Moreover, TQ has a role in the downregulation of the expression of STAT3-regulated gene products in gastric cancer in both in vivo and in vitro models [163]. Reports showed that TQ plays an essential role in the induction of apoptosis by decreasing the expression of antiapoptotic proteins, as well; it also significantly increased the expression of pro-apoptotic protein [164]. This process is mediated by the activation of caspases 8, 9, and 7 in a dose-dependent manner and increases the activity of PPAR-γ [165,166,167].

TQ prevents DNA damage caused by free radicals by scavenging the free radical activity [168,169,170]. TQ shows a significant effect in the decrease of expressions of CYP3A2 and CYP2C 11 enzymes [171]. TQ treatment showed activity in the reduction of CYp1A2, CYP 3A4, and CYp3A4 enzyme activity and the increase of phase II enzyme, GST. TQ has proven its role in tumor prevention through activation of antioxidant enzymes and its antioxidant activity [172]. TQ treatment illustrated a valuable role in the increase of the PTEN mRNA. Moreover, it has a pivotal role in the inhibition of breast cancer cell proliferation and induction of apoptosis via activation of the P53 pathway in MCF-7 cell line, the finding revealed that a time-dependent increase of PTEN occurs in cells treated with TQ as compared with the untreated cells [173]. TQ induces degradation of the tubulin subunit in the cells; it also inhibits the telomerase enzyme activity. Furthermore, it causes the suppression of androgen receptor expression and E2F-1 that is essential for the proliferation and viability of androgen-sensitive and androgen-independent prostate cancer cells [148].

### 2.7. Genistein

Genistein [4′,5,7-trihydroxyisoflavone or 5,7-dihydroxy-3-(4-hydroxyphenyl) chromen-4-one] is an isoflavonoid with a 15-carbon skeleton (Figure 7) and is classified as a phytoestrogen. It is found in food (especially legumes) in the glycosylated or free form. It is structurally similar to 17β-estradiol, which is the reason for its ability to bind to and modulate the activity of estrogen receptors [174].

It was isolated for the first time in the year 1899 from *Genista tinctoria*, hence it was named after the genus of this plant. However, it is the main secondary metabolite of the *Trifolium* species and in *Glycine max* (Soybean). In fact, soybean, soy-based foods, and soy-based drinks are the best sources of genistein. Lupin (*Lupinus perennis*) is also a legume that holds similar nutritional value to that of soybean in terms of genistein content. Other important legumes are broad beans and chick peas, which are known to contain significant amounts of genistein, although less than soybean and lupine [174,175]. The free form (aglycone form) of genistein is the pharmacologically active form and acts as anticancer, estrogenic, and antiosteoporetic agents. It can be extracted from its source through various means such as enzyme treatment and/or acid treatment followed by solvent extraction [176]. Other extraction methods have been reported in the literature including ultrasonication-assisted extraction [177], and supercritical fluid extraction with and without enzyme hydrolysis [178,179]. On the other hand, genistein can be chemically synthesized using conventional microwave ovens [174,176] or it can be biotechnologically synthesized by germinating the soybean seeds and enhancing its genistein content or by genetically manipulating non-legume crop such as rice (i.e., transgenic rice with high genistein content) [174].

Genistein exerts its anticancer effects by inducing apoptosis, decreasing proliferation, and inhibiting angiogenesis, as well as metastasis, which was illustrated in decreased tumor growth and development in hepatocellular cancer models of nude mice [180] and Wistar rats [181], as well as in gastric cancer model of Wistar rats [182]. Genistein role in prostate cancer was extensively studied in vivo in different animal models, such as Lobund-Wistar rat, (which is a unique rat model that spontaneously develops metastastic prostate cancer in 30% of its population), and in SCID mice transplanted with human prostate carcinoma cells (LNCaP, PC3, and DU-145). Some in vivo studies included normal rats to test for genistein toxic effect on prostate and its effect on the expression of androgen and estrogen receptor [183,184,185]. In addition, prostate cancer was addressed in vitro in several cell types (LNCaP, PC3, DU-145, PNT-1, PNT-2, and VeCaP) [184]. Genistein’s bioassay against several other cancer cell lines has been reported in 2017 by Estrela et al. [186], that included the following cancer cell lines and the resulting IC_50_ of genistein: breast carcinoma (MDA-MB-231 and T47D; IC_50_: 43 and 48 µM, respectively), colon carcinoma (HT29 and COLO201; IC_50_: 50 and 73 µM, respectively), lung carcinoma (A549 and NCI-H460; IC_50_: 64 and 47 µM, respectively), pancreas carcinoma (BxPC-3 and PANC-1; IC_50_: 79 and 87 µM, respectively), melanoma (MML-1 and SK-MEL-2; IC_50_: 42 and 36 µM, respectively), and glioblastoma (U87 and LN229; IC_50_: 55 and 44 µM, respectively).

Genistein is reported to inhibit cyclooxygenase-2 (COX-2) directly and indirectly by suppressing COX-2- stimulating factors like activated protein-1 (AP-1) and Nf-κB. COX-2 overexpression has been described in pancreatic, colon, breast, and lung cancer, and its inhibition has been correlated with decreased development of the cancerous tumor in the esophagus and in the colon [175]. Genistein inhibits CDK by upregulating p21, and it suppresses cyclin D1 which ultimately induce G2/M cell cycle arrest and decreases tumor cell progressions [175,184,186,187,188]. Genistein was reported to downregulate the expression levels of matrix metalloproteinase-2 (MMP-2) in glioblastoma, melanoma, breast, and prostate cancer cell lines. Matrix metalloproteinase (MMP) is the starting step in metastasis and angiogenesis cascade [175,189,190]. In addition, AP-1 is an angiogenic cytokine, which is inhibited by genistein, and, consequently, such an inhibitory effect will impede several targets, including Cyclin D1, MMP, VEGF, Bcl-2, uPA, and Bcl-XL [175]. Moreover, genistein can influence metastasis and induce apoptosis by inhibiting Akt, as well as NF-κB cascades, in PC3 cell lines and MDA-MB-231 breast cancer cell lines [175,191]. Further, genistein decreases phosphorylated-Akt in HT-29 colon cancer cells [192], in LNCaP prostate cancer cells [193], and in HeLa and CaSki cervical cancer cell lines [194], as well as in other cancer cell cultures [175].

Another important physiological process in which genistein is involved in is epigenetic modulation in a direct or indirect manner through estrogen receptor-dependent pathways [195]. Genistein was found to inhibit histone deacetylase (HDAC) enzymes, which are responsible of regulating histone acetylation of DNA [175], in MCF-7 and MDA-MB-468, and in immortalized but noncancer fibrocystic MCF10A breast cells at very low, dietary relevant concentrations [196]. A particular HDAC enzyme is HDAC-6 that is known to acetylate and activate heat shock protein (Hsp90). Basak et al. (2008) [197] reported that the increased ubiquitination of androgen receptors was due to the inhibition of Hsp90 chaperones in genistein-treated LNCaP prostate cancer cells. These results strongly support the hypothesis that genistein may be an effective chemopreventive agent for prostate cancer.

### 2.8. Parthenolide

Parthenolide is an important naturally occurring metabolite in the Asterceae family of medicinal plants [198], especially in *Tanacetum parthenium* (feverfew) [199]; however, it can be found in other species, including *Tanacetum vulgare* (tansy) and *Tanacetum larvatum* [200].

Parthenolide is primarily found in the plant shoots, or aerial parts, mainly flowers and leaves, and in minute amounts in the roots. However, commercially available parthenolide for research purposes has been extracted with more than 97% purity from *Tanacetum parthenium* leaves [201]. A conventional feverfew extraction was performed using chloroform and petroleum ether to extract parthenolide [202]. Later on, high-performance liquid chromatography (HPLC) gradient method was settled [203]. A number of other HPLC extraction methods were also reported [204,205]. Zhou et al. indicate that acetonitrile with 10% of water (*v*/*v*) using bottle stirring methods extracted the highest amount of parthenolide (930 mg/100 g raw material) from feverfew [205]. Furthermore, the isolation of parthenolide from feverfew with supercritical carbon dioxide extraction was reported [206,207,208]. Cretnik et al. compared the performance of conventional and high-pressure extraction techniques for separation of parthenolide from feverfew. They found that the best organic solvent for conventional extractions was acetonitrile, which extracted 350 mg parthenolide from 100 g of raw material, where the supercritical extraction followed by single-step separation showed that high operating parameters are required (600 bar, 60 °C) to achieve approximately the same amount of parthenolide isolated (328.8 mg/100 g raw material) as with acetonitrile extraction [209]. Additionally, studies revealed that the content of parthenolide in different parts of *Tanacetum parthenium* decreased during 18 months of storage [210].

Parthenolide is a sesquiterpene lactone with methylene-γ-lactone ring and epoxide group (Figure 8) which enables rapid interactions with biological sites [211]. In the past, Parthenolide was primarily used to treat migraine, fever, and rheumatoid arthritis, while recently, the studies find that parthenolide exerted anticancer effect in a variety of tumors, such as breast cancer, cholangiocarcinoma, pancreatic cancer, bladder cancer, prostate cancer, and leukemia [212]. Parthenolide has relatively poor pharmacological properties, derived from its low solubility in water and consequently reduced bioavailability, which limit its potential clinical use as an anticancer drug; however, a series of parthenolide derivatives were prepared to overcome this issue [213].

Parthenolide induces its anticancer effect through different mechanisms of action [214]. Its cytotoxic action could be a related to interruption of DNA replication by the highly reactive lactone ring, epoxide, and methylene groups [215]. Moreover, it mediated STAT3 inhibition inducing the expression of death receptors and, hence, an apoptotic pathway [216]. Furthermore, the molecular mechanism of parthenolide action are strongly associated with proapoptotic action through the activation of p53 and the increased production of reactive oxygen species (ROS) [199,210], along with reduced glutathione (GSH) depletion [214]. Duan et al. reported that parthenolide can target mitochondrial thioredoxin reductase to elicit ROS-mediated apoptosis [217]. Besides, parthenolide interferes with microtubule formation and preventing proliferation of malignant cells [218]. Parthenolide induces thrombopoiesis through the inhibitory activity of NF-κB and consequently render cancer cells prone to undergo apoptosis [219]. Additionally, parthenolide can impair focal adhesion kinase-dependent signaling pathways and, hence, the cell proliferation, survival, and motility [220]. Interestingly, studies showed that parthenolide specifically affects malignant but is harmless to normal cells [221].

Several studies illustrated the effectiveness of parthenolide as anticancer agent. Parthenolide suppressed tumor growth in a xenograft model of colorectal cancer by the induction of apoptosis [222], and it significantly reduced the development of colitis-associated colon cancer and histological acuteness in a murine model [223]. Kim et al. demonstrated the efficiency of parthenolide as anticancer agents against cholangiocarcinoma, intrahepatic bile duct carcinoma, since it can effectively induce apoptosis in four distinct cholangiocarcinoma cell lines [224]. Parthenolide showed potent chemopreventive potential against dimethylbenzene-anthracene (DMBA)-induced oral carcinogenesis, using hamster buccal as a model; oral administration of parthenolide completely prevented tumor formation and significantly reduced the nefarious histopathological changes [225]. Parthenolide exerted cytotoxic effects on breast cancer stem-like cells by inducing oxidative stress and necrosis [226]. Parthenolide exhibits antitumor properties and selectively induces radiosensitivity in mouse prostate cancer cell lines, while protecting primary prostate epithelial cell lines from radiation-induced damage [227].

Nakabayashi and Shimizu examined the effect of parthenolide on Glioblastoma, the most aggressive type of brain cancer, using a xenograft mouse model. They found that parthenolide significantly inhibited the growth of transplanted glioblastoma cells with respect to the control group [228]. Recent research indicates that parthenolide can enhance the antiproliferative effects of gemcitabine in pancreatic cancer cells [229]. Likewise, suberoylanilide hydroxamic acid, a histone deacetylase inhibitor, synergistically sensitized breast cancer cells to the cytotoxic effect of parthenolide [230].

### 2.9. Luteolin

Luteolin is a common flavonoid abundantly found in many plant species. It is predominantly present in fruits and vegetables, such as celery, chrysanthemum flowers, sweet bell peppers, carrots, onion leaves, broccoli, and parsley [231,232]. Moreover, it can be obtained from other plants, including *Sesbania grandifolra*, *Cajanus cajan*, *Apium graveolens*, *Platycodon grandiflorum*, *Mentha spicata*, and *Perilla frutescens* [233].

Different extraction techniques have been developed to extract luteolin from plants. Abidin et al. assessed the extraction of luteolin using maceration, heat reflux, and Soxhlet extraction. They concluded that the reflux technique using methanol as a solvent is better than the other extraction techniques [234]. Hydrodistillation was also used for flavonoids extraction including luteolin [235]. Besides, microwave-assisted extraction technique was employed in extraction of luteolin, and it was reported that this method is more efficient and produced higher extraction yield compared to maceration and heat reflux extraction methods [233]. Furthermore, ultrasonic-assisted method processed advantages on flavonoids-enrich extract, compared to heating extraction and microwaves-assisted extraction [236]. On other hand, Paula et al. reported the absence of luteolin in supercritical carbon dioxide extract and justified this by the hydrophilic nature of luteolin which makes luteolin a molecule insoluble in carbon dioxide [237]. Additionally, enzyme assisted extraction of luteolin was investigated by using three enzymes, namely cellulase, beta-glucosidase, and pectinase, and it was found that pectinase is more efficient than the other two enzymes, under the same conditions [238].

Luteolin is a polyphenolic flavone (3′,4′,5,7-tetrahydroxyl flavone) (Figure 9) [239]. Luteolin is often found in glycosylated form. It is composed of three benzene rings: A and B are solely benzene, while the third, the C ring, contains oxygen and carbon double bond at 2–3 position. The double bond and hydroxyl groups are important features in the structure of luteolin because these are associated with its biological activities [240,241]. Plants rich in luteolin have been used widely in Chinese traditional medicine. Luteolin exhibits multiple biological effects, such as anti-inflammation, anti-allergy, and anticancer, and it can act as an antioxidant [242].

Studies proved the therapeutic ability of luteolin against different types of cancer through multiple mechanisms. Luteolin acts as an anticancer agent by inhibition of cell proliferation, angiogenesis, metastasis, and induction of apoptosis through different mechanisms [242]. Luteolin activates both the extrinsic and intrinsic apoptosis pathways and increase the expression of the death receptor 5 [243]. Cai et al. demonstrated that luteolin can inhibit cell growth and induces G2 arrest and apoptotic cell death via activating JNK and inhibiting translocation of NF-κB [244]. Additionally, luteolin suppressed proliferation and survival of cancer cells by inhibition of angiogenesis through blocking activation of the VEGF receptor and its downstream molecule PI3K/Akt and PI3K/p70S6 kinase pathways [245]. Moreover, luteolin can suppress metastasis of cancerous cells by inhibitions of wide panel of receptor tyrosine-kinases activity such as Human epidermal growth-factor receptor 2 (HER-2), insulin-like growth factor (IGF) and epidermal growth-factor receptor (EGFR) [246]. Most flavonoids, including luteolin can act as antioxidants through different mechanisms [247]. First, Luteolin serves as a ROS scavenger where its structure satisfies the basic requirements for this activity: 3′,4′-hydroxylation, the presence of a double bond between carbons 2 and 3, and a carbonyl group on carbon 4. Second, luteolin inhibits ROS-generating oxidases such as xanthine oxidase activity. Third, luteolin may exert its antioxidant effect by protecting or enhancing endogenous antioxidants such as glutathione-*S*-transferase (GST), glutathione reductase (GR) and superoxide dismutase [242]. It goes without saying that antioxidants have a crucial role in cancer therapeutic strategy since oxidative stress involves in the pathophysiology of different types of cancer [248]. Over and above that, a recent study showed that gene-expression modulation is one of the mechanisms by which luteolin can induce its anticancer effect; it increased the expression of genes related to apoptosis and stress response within LC540 tumor Leydig cells [249].

Several studies confirmed the promising results of luteolin in the treatment of diverse types of cancers: breast, colon, pancreatic, prostate, oral, lung, skin, kidney, and ovarian cancer [231]. Luteolin effectively exerted a potent therapeutic effect on invasion and metastasis of breast cancer in human breast cancer cell lines MDA-MB-231 and BT5-49 [250]. Moreover, luteolin inhibits metastasis, cell migration, and viability of triple-negative breast cancer cells in xenograft metastasis mouse model [251]. Luteolin inhibited the proliferation, cell cycle progression and induced apoptosis of human colon cancer cell line LoVo suggesting the its potential as chemopreventive and chemotherapeutic agent against colon cancer [252,253]. Moreover, the evidence supported the activity of luteolin against non-small-cell lung cancer since it can inhibit cell proliferation and induced apoptosis of in both A549 and H460 cells [254,255,256]. Kasala et al. confirmed the antioxidant and antitumor activities of luteolin against benzo(a)pyrene-induced lung carcinogenesis in Swiss albino mice [257]. Tjioe et al. identified luteolin as a potent cytotoxic drug against oral squamous cell carcinoma with low toxicity and high efficiency [258]. It remains to mention that chemopreventive and chemotherapeutic effects of luteolin are also contributed to the synergistic effects with other anticancer therapies such as cyclophosphamide, doxorubicin, and NSAID such as celecoxib [259,260].

### 2.10. Quercetin

Quercetin is a naturally occurring polyphenolic flavonoid that is commonly found in different plant such as *Ginkgo biloba*, *Aesculus hippocastanum*, and *Hypericum perforatum* [261], as well as in fruits and vegetables including onions, apples, broccoli, berries, and green and red tea [262].

Different conventional methods were used for quercetin extraction. Simple cold extraction with ethyl acetate was reported as effective and fast method of isolation of crude quercetin with similar efficiency to hot ethyl acetate [263]. Additionally, quercetin was extracted from red and yellow onion skins, using supercritical carbon dioxide as a solvent [264]. Quercetin was successfully extracted with subcritical water extraction, an environmentally friendly process, with higher efficiency than those obtained by conventional extraction methods using ethanol, methanol, and water at boiling point [265]. Sharifi et al. demonstrated that the ultrasound assisted extraction was the most effective method for quercetin extraction compared to traditional methods including maceration, digestion and Soxhlet. The superiority of this method are shorter extraction times, the using of lower solvent amounts and higher extracted yield [266]. Zhang et al. found that extraction of quercetin using microwave is a rapid method with a higher yield and lower solvent consumption compared to Soxhlet and ultrasonic methods [267]. Moreover, ionic liquid-based pressurized liquid extraction procedure coupled with high performance liquid chromatography was developed and used to extract quercetin [268].

Quercetin, chemically known as 3,3′,4′,5,7-pentahydroxyflavone, is a flavonol type flavonoid [269] with *O*-dihydroxyl structure at the 3′ and 4′ positions in the B-ring, so-called catechol group (Figure 10), which is responsible for many biological activities of quercetin including its free radical scavenger effect [270]. Like other flavonoids, quercetin is commonly found in glycoside form [269]. Quercetin is a useful molecule with many pharmacological properties. It is well recognized as a neuroprotective, antiviral, antimicrobial, anti-inflammatory, hepatoprotective, cardiovascular, and reproductive system protective agent, as well as an antiobesity and anticancer agent [271].

Various cellular targets have been reported to be involved in the anticancer activity of quercetin. It has been reported that quercetin reduced the expression of epidermal growth factor receptor (EGFR), tyrosine kinases involved in the development of a wide variety of solid tumors, resulting in the inhibition of cell growth and the induction of apoptosis [272]. Quercetin also increased the expression of death receptor 5 (DR5) resulting in stimulation of tumor necrosis factor related apoptosis-inducing ligand (TRAIL) and subsequent cancerous cells apoptosis [273]. Moreover, Lee et al. reported that quercetin exerted its chemopreventive effect by direct targeting of Raf and MEK in Raf/MEK/ERK cascade which is important pathway in neoplastic transformation [274]. Studies also suggested that quercetin may induce apoptosis by direct activation of caspases cascade, increase the level of caspase-3 and -9, and then increase the expression of proapoptotic Bcl-2 family members and lower the levels of antiapoptotic Bcl-xL that contribute directly to the apoptotic process [275]. Srivastava et al. confirmed the interaction of quercetin with DNA directly as one of the mechanisms for inducing apoptosis in both cancer cell lines and tumor tissues, by activating the intrinsic pathway [276]. Like other flavonoids, quercetin can induce its anticancer activity through its antioxidants and radical scavenging properties [277,278,279].

Several studies documented the anticancer role of quercetin. Treatment of MCF-7 and MDA-MB-231 breast cancer cell lines with quercetin enhances apoptosis along with G1 phase arrest [280,281]. In addition, quercetin was found to enhance the chemosensitivity of breast cancer cells to doxorubicin via inhibiting cell proliferation and invasion, resulting improvement in cell apoptosis [282]. Further, a synergistic action of quercetin and curcumin was observed in triple-negative breast cancer cell lines by modulating tumor suppressor genes [283]. Quercetin can be a potent therapeutic agent for the treatment of colorectal cancer since it inhibits the survival and induces apoptosis in colon cancer cell lines, namely CT26, MC38, and CACO-2; it can also suppress colorectal lung metastasis in the mouse model [284,285]. Furthermore, studies revealed that treatment of mice with quercetin has a positive effect against 1,2-dimethyl hydrazine induced colon cancer approving its protective effects [286]. Likewise, quercetin significantly inhibits proliferation, promotes apoptosis, and induces cell cycle arrest within the G1 phase in pancreatic cancer cells [287]. Oršolić and Car demonstrated that treatment of male albino mice of Swiss strains with quercetin showed improvement in cisplatin inhibiting tumor growth activity besides a protective effect on cisplatin-induced DNA damage in normal cells [288]. Quercetin also showed promising results on human lung carcinoma A549 cells: It induced apoptosis, reduced the number of tumor cells, and suppressed cell invasion and migration [289]. Moreover, quercetin may be an effective chemopreventive and chemotherapeutic agent and could prevent cell propagation and colony formation of human bladder cancer cells [290,291]. Quercetin also considerably reduced the human malignant pleural mesothelioma cell viability and induced apoptotic cell death [292]. Ali and Dixit reported that oral administration of quercetin, at a concentration of 200 and 400 mg/kg body weight daily for 16 weeks, reduced the tumor size and the number of papillomas in skin tumor induced by croton oil in Swiss albino mouse [293]. More and more, evidence showed that combination quercetin treatment with X-irradiation increased the DNA damages and created common apoptotic cell death in ovarian cancer cell lines, namely OV2008 and SKOV3, compared to cells presented to quercetin or X-rays alone [294].

### 2.11. Paclitaxel

Paclitaxel (Taxol) was originally isolated from the bark of *Taxus brevifolia* (Pacific yew) as a result of a huge plant-screening program initiated by the National Cancer Institute in 1960s [295]. The needles and leaves of *Taxus baccata* (English yew) provided 10-deacetylbaccatin as precursor for semisynthesis of taxol [296,297]. Although Taxol is extracted in higher concentration from the bark of *Taxus brevifolia*, but bark harvesting destroys the tree and seriously threatens the very slow-growing yew tree population and has proved unsuitable for long term or large-scale production of Taxol [298]. Unfortunately, the demand for the drug exceeds its supply from natural sources. Therefore, the finding of alternatives sources is required. Although it was successfully prepared by total chemical synthesis, these methods are not commercially viable. Microbial fermentation is currently the most promising alternative for the production of taxol at an industrial level [299].

Different procedures have been developed to isolate paclitaxel from natural sources. The majority of ordinary solvent extraction processes reported in the literature have made use of methanol as the extraction solvent, where the accelerated solvent extraction method enhances the conventional extraction process using solvent at elevated temperatures [300,301]. On other hand, HPLC and mass spectroscopy analysis showed that acetone is the best solvent for the extraction of paclitaxel from *Taxus baccata* needles [298]. Pure paclitaxel from plant material was obtained using Acetone/water precipitation procedure [301]. The extraction of paclitaxel from the ground bark of *Taxus brevifolia* using supercritical carbon dioxide was also reported [302]. Additionally, a simple and rapid microwave-assisted extraction procedure was developed and optimized for the extraction of paclitaxel from the needles of yew trees *Taxus baccata* [303]. Tan et al. reported the validity of ultrasonic assisted extraction technique using methanol and magnetic ionic liquids as adjuvants to extract paclitaxel with good extraction yield and short time of extraction [304].

Paclitaxel is a diterpene alkaloid with a complex chemical structure because of its ring system and its many chiral centers (Figure 11) [305]. It has rigid ring system consists of four rings; amongst them, one is a cyclodecane and other an oxetane ring [306].

The main mechanism by which paclitaxel exerts its anticancer effect involves stabilization of cellular microtubules through binding β-tubulin subunit and inhibiting their depolymerization leading to block in the progress of mitotic division and prohibit cell division to ultimately cause apoptosis [307]. Besides, intra-tumoral concentrations of paclitaxel cause cell death due to chromosome miss-aggregation on multipolar spindles where the resultant daughter cells are aneuploid, and a portion of these die due to loss of one or more essential chromosomes [308]. Paclitaxel has also been found to target the mitochondria and inhibits the function of the apoptosis inhibitor protein B-cell Leukemia 2 (Bcl-2) [309].

Early study in 1980 indicated that paclitaxel was a potent inhibitor of cell replication and migration in mouse fibroblast cells [310]. Later on, many studies introduced this molecule as potent anticancer agents and as a first microtubule stabilizing Agent [311]. Now, paclitaxel has been approved by the FDA to be used alone or in combination with other anticancer agents for treatment of breast cancer, non-small-cell lung cancer and ovarian cancer. It is also can be used to treat many other cancers including head and neck, esophagus, bladder, endometrial, and cervical cancers [309,312].

### 2.12. Vincristine

Vincristine is a natural chemotherapeutic agent isolated mainly from the leaves of *Catharanthus roseus* (Madagascar periwinkle) [313,314]. The production rate of vincristine from its original source is very low which necessitate the presence of other sources [315]. Fortunately, vinblastine, another anticancer drug from the same plant presents at levels 1000 times higher than vincristine. Vinblastine is used as the parent drug to obtain vincristine through simple structural modifications [316]. It was reported that vincristine is produced by *Fusarium oxysporum*, an endophyte of *Catharanthus roseus* [317]. Additionally, biosynthesis of the anticancer vincristine in callus cultures of *Catharanthus roseus* is one of the promising alternatives [314].

Vincristine was simply extracted from leaves of *Catharanthus roseus* by soaking the plant material in the cold water/0.1% methanolic HCl (1:1 *v*/*v*) for overnight [318]. Charcoal column was reported as simple and reliable method for isolation of highly purified form of vincristine and vinblastine from the dried plant of *Catharanthus roseus* [319]. Furthermore, Supercritical fluid extraction using carbon dioxide with basic modifier was used to extract vinblastine and vincristine from the aerial portions of *Catharanthus roseus* [320]. Karimi and Raofie reported optimized supercritical fluid extraction method of vincristine from *Catharanthus roseus*, using ethanol as co-solvent for carbon dioxide [321]. Moreover, an improved method termed negative-pressure cavitation extraction followed by reverse phase high-performance liquid chromatography was developed for the extraction and quantification of vincristine from *Catharanthus roseus* leaves. The yield of this method is comparable to that obtained by the well-known ultrasound assisted extraction method and significantly higher than that obtained by maceration extraction and heat reflux extraction [322]. Recently, Santana-Viera et al. isolated vincristine from marine fish using microwave assisted extraction technique [323].

Vincristine is bis-indole terpenoid alkaloid (Figure 12) [324]. The monomeric precursor molecules are vindoline and catharanthine. Vincristine is the oxidized form of vinblastine [322].

Vincristine induced its anticancer effects mainly by inhibition of polymerization of the microtubules through binding with the tubulin. This producing an arrest in G2/M phase and inducing apoptosis [325]. It is also known that vincristine is a potent inhibitor of topoisomerase II [326]. Additionally, vincristine shows high affinity to chromatin; binding of vincristine alters chromatin structure that perturbs histone-DNA interaction, and possibly removal/displacement of the histones from DNA is occurs, resulting in increasing its cytotoxic effect [327]. Vincristine was initially discovered as anticancer agent in 1963 when it was capable of curing mice transplantable leukemia P-1534 [328]. Now, vincristine is a potent and widely used anticancer agent, particularly for childhood and adult hematologic malignancies and solid tumors, including sarcomas, Hodgkin’s disease, non-Hodgkin’s lymphoma, Wilms’ tumor, and neuroblastoma [329,330].

### 2.13. Bromelain

Bromelain is an extract of pineapple (*Ananas comosus*) that contains a mixture of proteases and non-protease components [331]. Bromelain is abundant in stem and fruit of pineapple plant, and it can also be isolated in small amount from other parts, such as the core, leaves, and peel [332]. Bromelain is accumulated in various parts of the plant to different level, and its properties vary based on its source [333]. Assays for the individual protease components of bromelain have recently been established raising the possibility of standardizing bromelain preparations [334]. Bromelain is sold in health food stores as a nutritional supplement to promote digestion and wound healing, and as an anti-inflammatory agent [335].

Bromelain can be easily extracted from pineapple juice by ultrafiltration; however, fruit bromelain (FBM) is not commercially available, due to being different from stem Bromelain (SBM) [335,336]. The traditional methods for bromelain isolation have been through microfiltration and ultrafiltration followed by chemical precipitation using ammonium sulfate and then ultracentrifugation [337]. Extraction and purification of the bromelain was reported through aqueous two-phase system using poly ethylene oxide (PEO)-poly propylene oxide (PPO)-poly ethylene oxide (PEO) block copolymers [338]. Additionally, several studies demonstrated the purification of bromelain by a single step of polymer/salt aqueous biphasic system [339,340,341]. Furthermore, reverse micellar extraction can be successfully employed for the selective extraction and separation of bromelain [342]. Campos et al. reported that bromelain can be extracted by formation of non-soluble complexes with carrageenan, natural polysaccharides, as precipitant agents. Moreover, it was proved that bromelain maintained its biological activity through this precipitation process, since carrageenan also acted as enzyme stabilizer [343]. Ion exchange chromatography also applied for extraction and purification of bromelain [344,345]. Devakate et al. found that the purity of bromelain obtained by chromatography was 3.3 times compared to that obtained by precipitation [345]. Additionally, membrane technology was successfully employed for the selective extraction and separation of bromelain from the pineapple through two-stage ceramic ultrafiltration [346]. Besides, affinity membranes have been used to extract bromelain. Zhang et al. prepared an affinity membrane chemically modified with chitosan as a composite bilayer membrane, which in turn was covalently attached to Cibacron Blue, a stain specific to bromelain, this membrane showed high bromelain adsorption capacity [347].

Bromelain is a crude aqueous extract contains a mixture of different proteases; however, it is rich in cysteine proteases. It contains non-protease components such as phosphatase, glucosidases, peroxidases, cellulases, and glycoprotein [333,348].

Recent studies have shown that bromelain has the capacity to modify key pathways that involve in cancer treatment [349]. Bromelain inhibit the growth of cancer cells by increasing the expression of p53 and Bax activators genes of apoptosis in cancerous cells [350]. Bromelain inhibited the proliferation of cancerous cells and induced apoptosis via activating both caspase dependent and independent pathways [351]. Bromelain diminished the expression of the cell cycle regulatory proteins cyclin A, cyclin B, and cyclin D, resulting in G1 arrest [352]. Bhui et al. stated that bromelain exerted its antitumor activity through inhibition extracellular signal regulated protein kinase (ERK1/2) and p38 mitogen-activated protein kinase (MAPK) besides the decrease in Cox-2 expression and inhibition of NF-κB pathway [353]. Additionally, bromelain showed an antiangiogenic effect by interfering with VEGF [354,355]. Furthermore, bromelain was shown to stimulate ROS, and this would have a direct impact on the modulation of signaling in cancer cells led to tumor cell killing properties [334].

Many existing evidences indicates that bromelain can be a promising candidate for cancer treatments. Commercially available bromelain exerted strong effects towards MCF-7 breast cancer cells; it showed an inhibitory effect against proliferation of MCF-7 with IC_50_ values of 5.13 µg/mL, compared to taxol with IC_50_ value of 0.063 µg/mL, and the microscopic observation of bromelain-treated MCF-7 cells demonstrated detachment [356]. It also enhanced apoptosis in breast cancer cells with upregulation of c-Jun *N*-terminal kinase and p38 kinase [357]. Bromelain also exhibited inhibitory effects against both human epidermoid carcinoma-A431 and melanoma-A375 cell lines and caused depletion of intracellular glutathione and generation of reactive oxygen-species followed by mitochondrial membrane depolarization which led to cell cycle arrest at G2/M phase [358]. In vivo antitumor effect of bromelain using mice injected with different panel of leukemia cells was assessed; bromelain showed significant increase in survival index especially with mice bearing Ehrlich ascitic tumor, and this effect was superior to that obtained using 5-fluorouracil (318% versus 263%) [359]. Additionaly, bromelain showed tumor inhibitory effects in chemically induced mouse skin papillomas; topical application of bromelain delayed the onset of tumorigenesis and reduced the cumulative number of tumors, tumor volume, and the average number of tumors/mouse [360]. Moreover, Bromelain showed synergistic effects with other anticancer agents. Mohamad et al. confirmed that bromelain treatment could potentiate the antitumor effect of cisplatin on triple-negative breast cancer 4T1 cells through modulating the tumor environmental inflammation [361]. Furthermore, the presence of peroxidase enhances the biological efficiency of bromelain; a study by Debnath et al. evaluated the anticancer effect of bromelain in presence or absence of peroxidase in different cancer cell lines, and it established that the fresh pineapple has higher apoptosis potential due to the presence of the peroxidase [362].

### 2.14. Boswellic Acid

Boswellic acids are phytochemicals obtained from the gum resin of the *Boswellia* species which belong to the family Burseraaceae [363]. Boswellic acids are the major constituents of *Boswellia serrata* commonly known as Salai guggal, white guggal and Indian olibanum [364].

Conventional extraction methods such as Soxhlet extraction, percolation, and solvent extraction have been used for extraction of boswellic acids from *Boswellia serrata* gum resin. In the solvent extraction process, variety of solvents such as hexane, ethyl acetate, ethanol, methanol, acetone, and petroleum ether are mostly used [365,366,367,368]. Moreover, Sharma et al. developed high performance liquid chromatography method for qualitative and quantitative analysis of boswellic acids extracted from *Boswellia serrata* using different techniques [369]. It was reported to use ultrasound assisted extraction to extract boswellic acids with lesser extraction time compared to Soxhlet extraction [370]. Niphadkar et al. introduced three phase partitioning technique as an alternative and simple method for extraction of boswellic acids from *Boswellia serrata* plant oleo gum resin. It had high extraction yield, compared to conventional Soxhlet extraction, batch extraction, and novel UAE methods with low solvent consumption [371].

Boswellic acids are pentacyclic triterpenoids belong to ursane group [372]. Six derivatives of boswellic acids have been identified in this plant, namely α- and β-boswellic acid, acetyl-α- and acetyl-β-boswellic acid (ABA), 11-keto-β-boswellic acid (KBA), and 3-*O*-acetyl-11 keto-β-boswellic acid (AKBA) [373]. The two most potent anticancer boswellic acids of *Boswellia* are acetyl-11-keto-β-boswellic acid (AKBA) (Figure 13) and 11-keto-β-boswellic acid (KBA) [374].

It is now well established that boswellic acids are multitargeting agents. They can modulate several molecular targets, including enzymes, growth factors, kinases, transcription factors, receptors, and others related to the survival and proliferation of cells [375]. Studies revealed that boswellic acids induced their antitumoral activity through inhibition of topoisomerases I and II leading to apoptosis in different cell lines [376,377]. Additionally, Liu et al. reported that AKBA treated cancer cells exhibited cell arrest at G1 phase through downregulation of G1 phase cyclins and cyclin-dependent kinases (CDK). They also found that the G1 phase arrest induced by AKBA was dependent upon the expression of CDK inhibitor p21 [378]. Boswellic acids strongly induced apoptosis accompanied by activation of caspase-3, -8, and -9, resulting in expression of DR4 and DR5 [379,380,381]. Besides, AKBA potently suppressed tumor growth through inhibition of angiogenesis by targeting vascular endothelial growth factor (VEGFR2) signaling pathway [382]. AKBA also prohibited the phosphorylation of extracellular signal regulated kinase-1 and -2 (Erk-1/2) and impaired the motility of cancer cells; the Erk pathway plays a crucial role in signal transduction and tumorigenesis [383]. In accordance with this, Li et al. recently confirmed that AKBA suppressed the growth of glioblastoma cells by inhibiting autophagy through regulating the ERK and P53 signaling pathways [384]. Moreover, AKBA potentiated the apoptosis induced by cytokines and chemotherapeutic agents, suppressed TNF-induced invasion through inhibition of NF-κB regulated gene expression [385]. Latterly, Wang et al. suggested that induction of premature senescence by AKBA through DNA damage response accompanied by impairment of DNA repair genes as a novel mechanism contributing to AKBA growth suppression in hepatocellular carcinoma [386].

A number of researchers have reported that pentacyclic triterpenes of Boswellia, boswellic acids, are one of the most promising anticancer agents. Syrovets et al. proved that boswellic acids inhibited proliferation and induced cell death in chemoresistant androgen independent PC-3 human prostate cancer cells [387]. AKBA also showed antiproliferative effects against different colon cancer cell lines, namely HT-29, HCT-116, and LS174T. It also induced arrest in these cells at G1 phase [378]. Furthermore, it was demonstrated that although both curcumin and AKBA treatments can suppress tumor growth in a mouse xenograft model of colon cancer, the combined treatment resulted in synergistic tumor suppression [388]. Boswellic acids showed antitumor activity against human leukemia HL-60 cells; they inhibited the synthesis of DNA, RNA, and protein in in a dose-dependent manner [389]. Recently, Lv et al. demonstrated the anticancer effect of acetyl-11-keto-β-boswellic acid on human non-small-cell lung cancer cell lines, namely A549, H460, and H1299, via cell cycle arrest at the G0/G1 phase, apoptosis induction, and autophagy suppression [390]. Moreover, Xue et al. proved the potential use of AKBA to reverse multidrug resistance in human ileocecal adenocarcinoma; the study showed that cytotoxicity of vincristine increased drastically in vincristine resistance cells, HCT-8/VCR [391]. Boswellic acids also showed synergistic effect with cisplatin against hepatocellular carcinoma that was induced by orally administration of diethyl nitrosamine in rats [392]. Table 1 summaries the previously discussed natural products, their analogues, and the mechanisms of action as anticancer agents.

## 3. Drug Formulation and Clinical Studies

### 3.1. Curcumin

Although currently under investigation in human clinical trials, instability and low bioavailability due to low aqueous solubility has hampered the desired therapeutic use of curcumin. This dictates using targeted delivery approaches such as nanotechnology approaches for better medical application. Most of the formulations focus on enhancing the bioavailability and solubility of curcumin and protecting it from inactivation. Some of them are targeted for sustained circulation and retention in the body, while others focus on targeted delivery and intracellular release [411]. The following part discusses the main delivery strategies used to enhance the bioavailability, solubility and stability of curcumin.

#### 3.1.1. Nanoparticles

Nanoparticles are widely used drug delivery systems to increase the aqueous dispersibility of hydrophobic drugs. Nanoparticulate formulations of curcumin have been shown to increase in vivo oral bioavailability of the compound by at least nine-fold [412]. Polymeric, solid lipid, magnetic, gold, and albumin-based nanoparticles are examples of nanoparticles that are extensively used to improve curcumin therapeutic applications.

Polymeric nanoparticles are able to circulate in the blood for a long time [413]. N-isopropylacrylamide (NIPAAM), polyvinyl alcohol (PVA), poly(lactic-co-glycolic acid) (PLGA) [414,415], polyethylene glycol monoacrylate [NIPAAM (VP/PEG A)], *N*-vinyl-2-pyrrolidone, silk fibroin, modified starch, chitosan [416], silica nanoparticles [417], casein nanoparticles (Sahu, Kasoju et al., 2008), and Eudragit R E100 cationic copolymer [418] are examples of nanoformulation polymers. 

Other nanoparticle formulations of curcumin include glycerol monooleate and pluronic F127 curcumin loaded nanoparticles [419], and curcumin-loaded zinc oxide (ZnO) nanoparticles [420].

#### 3.1.2. Liposomes

Liposomes (characterized by the presence of one or more phospholipid bilayers surrounding an aqueous inner space) are considered as ideal delivery systems for biologically active substances due to their high stability, high biocompatibility and biodegradability, high solubility, low toxicity, targeting specific cells, controlled distribution, and easy preparation [421].

Many studies have shown that liposomal drugs accumulate mainly in the liver, spleen, lungs, bone marrow, or other tissues and organs, which helps decrease the side effects. They were shown to be the most suitable vehicle to treat various cancer diseases [422].

#### 3.1.3. Adjuvants

Adjuvants are compounds used to inhibit the metabolic inactivation or accelerated clearance of curcumin. Piperine is a well-known bioavailability-enhancing adjuvant that inhibits hepatic and intestinal glucuronidation enzymes, thereby improving intestinal absorption and systemic bioavailability of curcumin [423].

Other agents such as quercetin, naringenin, genistein, epigallocatechin-3-gallate and eugenol have been reported to have synergistic effects when used in combination with curcumin [424].

Another strategy to enhance the solubility of curcumin was by utilizing the solubilizing properties of rubusoside (RUB). The solubility increased linearly from 61 µg/mL to 2.318 mg/mL [425] and the RUB-solubilized curcumin was stable in physiological conditions when reconstituted. Additionally, the RUB-solubilized curcumin showed efficacy against human colon, breast, and pancreatic cancer cell lines [425].

#### 3.1.4. Micelles and Phospholipid Complexes

These complexes refer to a group of amphiphilic surfactant molecules which aggregate into spherical vesicles in water. They increase bioavailability of drugs by enhancing their solubility and gastrointestinal absorption.

Polymeric micelles of curcumin (Cur-M) introduced by Liu and co-workers were successful in halting the growth of breast tumors and spontaneous pulmonary metastasis [426]. Curcumin-poly(ethylene glycol) methyl ether (MPEG-PCL) micelles were useful in pulmonary carcinoma treatment [427]. In another study, small sized curcumin loaded micelles were suggested to have better cytotoxic effect on the human colon carcinoma cells compared to larger micelles [428]. Moreover, curcumin loaded into zein-poly(sulfobetaine methacrylate) (zein-PSBMA) micelles had considerably better stability, cellular uptake and cytotoxicity to cancer cells compared with the free curcumin [429].

#### 3.1.5. Conjugates

Conjugates are complexes formed by joining two or more molecules mainly by a covalent bond to improve the solubility and stability of drugs. Curcumin conjugates with hyaluronic acid [430], piperic acid and glycine [431], glutaric acid [12], gold nanoparticle-PVP [432], methoxy poly(ethylene glycol) (mPEG), and PLA [433,434] are examples. Curcumin has also been coupled to peptide/protein carriers such as beta-casein, an amphiphilic polypeptide to form micelles [435].

#### 3.1.6. Cyclodextrins

Cyclodextrins are carrier systems consisting of cyclic oligosaccharides with a hydrophilic outer surface and a lipophilic cavity, which can solubilize hydrophobic compounds such as curcumin [433]. They are composed of six (α-), seven (β-), or eight (γ-) d-glucopyranose units linked through α-1,4-glycosidic bonds to form macrocycles [436]. Derivatives of β-CD and γ-CD are widely used because of their low price, relatively easy synthesis, and adaptability. β-CD-mediated curcumin drug delivery systems exhibit enhanced distribution and therapeutic value of curcumin in prostate cancer cells compared to unformulated curcumin [411].

#### 3.1.7. Solid Dispersions

Solid dispersions are dispersions of a poorly water-soluble compound (dissolved in either amorphous or semi crystalline form) in an inert matrix to enhance its solubility [433]. Examples include the solid dispersions of 2-hydroxypropyl-β-cyclodextrin (HP-β-CD)-curcumin co-precipitates [437] and curcumin-polyethylene glycol-15-hydroxystearate solid dispersions [438].

#### 3.1.8. Nanospheres and Microcapsules

Nanospheres are solid matrix particles in which the drug is mixed, while microcapsules have an internal core and outer polymeric shell. Surfactant-free curcumin nanospheres (with an anticancer effect on breast cancer and osteosarcoma) [439], curcumin encapsulated PLGA nanospheres (with a potential clinical application in prostate cancer) [440], dimethyl curcumin encapsulated PLGA nanospheres (ASC-J9) (used on breast cancer cells) [441], curcumin-poly(ethylene glycol)-poly (lactic acid) (PEG-PLA) nanospheres (effective against HeLa and MDA-MB-231 cancer cells) [442] are examples for nanospheres. Encapsulation of curcumin in microcapsules containing a solid lipid nanoparticle and a mesoporous silica shell [443], and curcumin-polylactic acid (PLA)-based microcapsules [444] are examples for microcapsules.

#### 3.1.9. Miscellaneous Nanoformulations

Anuchapreeda and co-workers prepared a curcumin nanoemulsion based on soybean oil [445]. In another study, curcumin-loaded lipid-core poly (ε-caprolactone) nanocapsules coated with polysorbate 80 (C-LNCs) were developed. They exhibited 100% encapsulation efficiency [446].

Nanogels, yeast cells, metallo-complexes, and nanodisks are other types of formulations to enhance the biological activity of curcumin [411]. A nanogel is a nanoparticle system composed of a hydrogel crosslinked to a polymer. This structure offers a strong base for drug delivery and release [447]. Curcumin-loaded hydrogel nanoparticles formed by combining hydroxypropyl methylcellulose and polyvinyl pyrrolidone [448], curcumin-loaded gold nanoparticles–chitosan nanogels [449], and curcumin delivered as self-assembled capsules with carboxymethyl cellulose and casein nanogels with folic acid and casein [450] are examples.

Nanodisks which are disk-shaped, apolipoprotein-stabilized, and self-assembled systems, are used to enhance the solubility and target the release of curcumin [411].

Yeast cell-loaded curcumin formulation was shown to protect curcumin from environmental factors such as light, humidity, and heat [451]. Curcumin complexed with palladium (II) exhibited a strong anticancer effect to MCF-7, HeLa, and A549 cancer cells [452].

The phytosomal formulation of curcumin, a complex of curcumin with phosphatidylcholine, has been shown to improve curcumin bioavailability [453]. The phytosomal formulation of curcumin is safe, shown to enhance the oral bioavailability and stability against metabolism, and is efficacious against several human diseases including cancer [454]. In vivo and human studies suggested that this formulation has good properties for clinical use [455].

Numerous clinical trials have studied the pharmacokinetic profile, safety, and effectiveness of curcumin to different diseases, including cancer. Some promising positive results showed that curcumin could arrest or even eliminate the growth of cancer cells [411].

The free form and nanoformulations of curcumin have been under investigation in human clinical trials for many years and curcumin has shown clinical benefits against various types of cancer including multiple myeloma, colorectal, pancreatic, and breast cancers. Curcumin is mostly administered orally as capsules in high doses due to its low bioavailability. Studies revealed that doses up to 12 g per day have no toxicities [433].

#### 3.1.10. Curcumin Clinical Studies

In a controlled semi-quantitative clinical study, Meriva (curcumin phytosomes) was evaluated to assess its potential to alleviate the side effects of cancer chemotherapy and radiotherapy. Additionally, this formulation was tested as an adjunct to chemotherapy in a group of patients with solid tumors (administered 1500 mg/day in three divided doses) for six weeks. It significantly enhanced quality of life of patients and suppressed systemic inflammation [456].

In a pilot study to assess the efficacy of Meriva in benign prostatic hyperplasia (BPH) (administered 1000 mg/day in two divided doses), there were improvements in all items of the International Prostate Symptom Score, with a better efficacy and without side effect [457].

The safety and anticancer activity of curcumin in human participants with colon cancer were demonstrated in a clinical study conducted by Shehzad and co-workers [458]. In a phase I clinical trial, in which curcuma extract administered orally to patients with colorectal cancer at doses of up to 2.2 g daily (equivalent to 180 mg of curcumin) for several months, curcumin was shown to accumulate at the colorectum, and achieve the effective therapeutic concentration, which illustrated the potential of curcumin to cure colorectal cancer [459].

In a phase II trial, twenty-five patients with advanced pancreatic cancer received 8 g/day of curcumin capsules, with restaging every eight weeks [460]. No toxicity was observed when the drug levels peaked at 22 to 41 ng/mL. This study demonstrated that despite its limited absorption, oral curcumin showed biological activity and was safe enough in some patients with pancreatic cancer. Moreover, the expressions of NF-κB, COX-2, phosphorylated signal transducer, and activator of transcription 3 (which were higher in patients compared to healthy volunteers), were found to be decreased in the peripheral blood mononuclear cells in most patients.

In a phase I trial, docetaxel plus curcumin given to patients with advanced breast cancer with dose escalation, showed some improvements such as biological and clinical responses in most patients [461]. In another study, the efficacy of the co-administration of curcumin and quercetin to regress adenomas in patients with familial adenomatous polyposis (FAP) (an autosomal dominant disorder characterized by cancer of the colon and rectum) was investigated. FAP patients with prior colectomy were orally given curcumin a dose of 480 mg and quercetin 20 mg three times daily for six months. The number and size of these polyps that will become cancerous were significantly reduced [462].

In another clinical study to evaluate the expression of prostate-specific antigen (PSA) which reflect the development of prostate cancer, 43 healthy participants received a supplement containing 40 mg isoflavones (66% daidzein, 24% glycitin, and 10% genistin) and 100 mg curcumin, and 42 volunteers received placebo daily in a double blind study for six months. The levels of PSA decreased in some volunteers and curcumin was well tolerated. This indicates that the combination may have therapeutic advantages in prostate cancer [463].

A randomized double-blind placebo-controlled parallel-group comparative clinical study was conducted to assess the efficacy and safety of an intravenous infusion of curcumin in combination with paclitaxel in a group of 150 women with advanced breast cancer. The patients were followed up for three months. This study concluded that treatment with curcumin in combination with paclitaxel was superior to the paclitaxel-placebo combination [464].

In a phase I clinical trial, the safety, pharmacokinetics and tolerability of intravenous liposomal curcumin was assessed in healthy volunteers. The short-term administration of this formulation was safe for up to a dose of 120 mg/m^2^ [465].

### 3.2. Resveratrol

Although resveratrol (RES) was demonstrated to have a promising therapeutic potential as confirmed by in vitro studies, animal and clinical trials have shown less promising results because of the extremely low bioavailability of oral RES [466]. At present, RES is only administered orally. Alternatively, using other routes of administration or delivery systems to avoid first-pass metabolism will increase the bioavailability of RES and the concentrations at the active sites. This part of the paper focuses on the alternative formulations and delivery systems associated with the potential use of RES as an anticancer agent.

#### 3.2.1. Oral Transmucosal Administration

Oral transmucosal administration of resveratrol using ribose lozenges achieved higher and faster blood concentrations of RES compared with oral administration [467]. In another approach related to buccal delivery, RES was formulated as cyclodextrin-based nanosponges. This delivery system displayed improved release, stability, and accumulation of RES in rabbit mucosa [468].

#### 3.2.2. Metabolites

Some of RES metabolites such as sulfate metabolites have been shown to regenerate and produce biologically active concentrations of RES in plasma and tissues [469]. In relation to controlling RES metabolism, the bioavailability of RES was enhanced by administering RES in combination with quercetin which inhibits its glucuronidation and sulfation [470].

#### 3.2.3. Novel Formulations

A product consisting of red grape cells (RGC) in which RES with one hexose moiety was the main polyphenol, revealed a high bioavailability and solubility in body fluids, and rapid gastrointestinal absorption compared with RES alone. This glycosylated structure of RES provides more stability and resistance to enzymatic metabolism [471].

#### 3.2.4. Dose Manipulation

In an attempt to enhance the oral bioavailability of RES, saturating the enzymes responsible for its metabolism with dose-escalation of RES was found to increase the oral bioavailability of RES with linear pharmacokinetics [472].

#### 3.2.5. Naturally Occurring RES Analogues

Some naturally occurring analogues of RES such as RES trimethyl ether (*trans*-3,5,4′-trimethoxystilbene, RTE) has been found to have better metabolic stability than RES. This is attributed to the complete methoxylation of the hydroxyl groups of RES. The relatively lower aqueous solubility which determines the oral administration of RTE could be improved by drug delivery systems such as cyclodextrin (randomly methylated-β-cyclodextrin (RM-β-CD) [473].

Pterostilbene (*trans*-3,5-dimethoxy-4′-hydroxystilbene, PTS), another analogue of RES, has been shown to exhibit more favourable pharmacokinetics compared with RES. PTS possesses less hydroxyl groups than RES, which makes it less susceptible to metabolism. PTS undergoes extensive distribution to major drug target organs, such as the kidneys, liver, heart, brain, and lungs [474].

Oxyresveratrol (*trans*-3,5,2′,4′-tetrahydroxystilbene, OXY), another analogue of RES, possesses an additional phenolic hydroxyl group which results in better water solubility than RES. The pharmacokinetic profile of OXY and RES in Sprague Dawley rats and their pharmacokinetic profiles were found comparable [472]. Furthermore, OXY demonstrated good oral bioavailability, faster absorption, and much slower clearance compared with RES.

Isorhapontigenin (*trans*-3,5,4′-trihydroxy-3′-methoxystilbene, ISO), a methoxylated analogue of RES, has been shown to be more orally bioavailable than RES, approximately by 50% [475].

*Trans*-4-4′-dihydrostilbene (DHS) displayed promising anticancer activity in preclinical studies. As the major barrier with DHS was its aqueous solubility, the aqueous solubility of DHS was overcome by solubilizing it with hydroxypropyl-β-cyclodextrin. This resulted in an improved pharmacokinetic profile compared with RES alone [473,476].

Some RES derivatives such as gnetin-C, a naturally occurring RES dimer, were found to have better anticancer properties and pharmacokinetic profile than RES against acute myeloid leukemia (AML) [477]. A combination of curcumin and RES exhibited a synergistic chemopreventive response in lung carcinogenesis in mice [478]. 

In another approach, RES was reformulated as acyl-glucosyl derivatives which resist absorption and are able to be steadily de-conjugated in the gastrointestinal tract to provide an effective dose of free RES to the colonic mucosa [479].

#### 3.2.6. Nanotechnology

Solid lipid nanoparticles (SLNs) [480], gold nanoparticles [481], cationic chitosan- and anionic alginate-coated poly d,l-lactide-coglycolide nanoparticles [482], and nanocores and nanocapsules [483,484] are nanotechnology-based delivery systems for RES. Nanotechnology-based delivery systems for RES also include polymer-based nanocarriers, electrospun nanofibers, lipid-based nanocarriers, CDs, and other nanocarriers [485].

#### 3.2.7. Solid Lipid (SLNs), Gold, and Chitosan Nanoparticles

SLNs formulations of RES have been shown to significantly increase the oral bioavailability of RES, protecting the incorporated RES from rapid metabolism in addition to the controlled release properties. Glyceryl behenate SLNs were used to deliver RES to the brain as a strategy to treat glioma [486].

In addition to enhanced bioavailability and cellular uptake, RES conjugated to gold nanoparticles have been shown to enhance anticancer activity in comparison with the free form of RES [487].

Cationic chitosan- and anionic alginate-coated poly d,l-lactide-coglycolide nanoparticles have been found to increase stability of RES and improve drug loading and controlled release mechanisms. Moreover, these nanoparticles have been found to efficiently prevent or suppress cancers after intravenous or topical administration [482].

Polymer-based nanocarriers are composed of natural or synthetic polymers. Examples for natural polymers include polysaccharides such as chitosan (CS) [488,489] and proteins such as gelatine [490], Zein [491], fibroin [492], and albumin [493].

Synthetic polymers include homopolymers such as poly (lactic-co-glycolic acid) (PLGA) [494], PLGA coupled with a galactose ligand (*N*-oleoyl-d-galactosamine) [495], transferrin (Tf) [496], poly (lactic acid) (PLA), poly (ε-caprolactone) (PCL), and copolymers such as poly (acrylic acid)-poly (methacrylic acid) commercially known as Eudragits [497].

RES delivery system using PEG and PLA exhibited anticancer effects on in vitro and in vivo colon cancer models [498]. Additionally, PEG-PLA conjugated to transferrin (Tf) was used to target glioma [496].

Moreover, encapsulation of RES in Eudragit E100-PVA NPs significantly improved the RES release profile and the RES antioxidant and anti-inflammatory activity, accounting for its in vivo hepatoprotective effect [497].

#### 3.2.8. RES Nanocores and Nanocapsules

Nanocores of RES using polyvinylpyrrolidone 17 PF (PVP K17) as the stabilizer and poloxamer 188 (F188) as the surfactant were developed by Hao and co-workers [483].RES nanocapsules containing RES nanocores coated with multilayered polyelectrolyte shells showed enhanced RES bioavailability compared to the free RES carboxymethylcellulose suspension according to in vivo studies using oral administration [484].

#### 3.2.9. Electrospun Nanofibers

The high surface area of these carriers makes them suitable for the controlled delivery of highly unstable and volatile compounds [499]. Additionally, these carriers are able to mimic the extracellular matrix and facilitate the cell adhesion, proliferation, and differentiation processes.

#### 3.2.10. Lipid-Based Nanocarriers

Lipid-based nanocarriers are composed primarily of digestible lipids which promote the intestinal absorption of drugs. Lipid-based nanocarriers include nanoemulsions and liposomes [466]. Nanoemulsions are colloidal particulate systems which provide high surface area and great stability [500]. Self-nanoemulsifying delivery systems (SNEDSs) which are preconcentrates or anhydrous isoforms of nanoemulsions, are able to pass the first-pass hepatic portal route and mediate lymphatic transport of lipophilic drugs [501]. The SNEDSs delivery system of RES was able to inhibit growth in MCF-7 breast cancer cells [502]. Recently, gum arabic was used with whey protein as an emulsifier to enhance emulsification and the storage stability of the RES nanoemulsion [503].

A Pickering emulsion where stabilizers such as quinoa starch granules are particles, is another type of emulsion. RES loaded with this type of emulsion was more stable compared to the emulsion stabilized with Tween 20 [504].

RES liposomes composed of lecithin and cholesterol were found to significantly accelerate the in vivo RES absorption [484].

#### 3.2.11. Cyclodextrins

As the structure of cyclodextrins has three hydroxyl groups, RES may be attached to the inner hydrophobic cavity of the semi-synthetic CDs, β-CDs [505]. This increases the amount of RES in aqueous solution and delays the oxidation of RES [506].

In another study, inclusion complexes using HP-β-CDs and randomly methylated-β-cyclodextrins (RM-β-CDs) were prepared to enhance RES solubility and bioavailability [507].

#### 3.2.12. Additional Nanocarriers

In nanocrystals, drugs are organized in a crystalline structure on the nanoscale [508]. RES nanocrystals, prepared using different combinations and proportions of d-α-tocopherol polyethylene glycol 400 succinate (TPGS), lecithin, and Pluronic F127 (PF127), exhibited improved RES solubility, bioavailability, and uptake across the intestinal barrier when orally administered to Sprague Dawley rats [509].

Another nancarrier is Niosomes which have lamellar structures formed by self-assembly of non-ionic surfactants in combination with fatty alcohols. RES niosomes for oral administration were produced by the thin film hydration method [510]. In another study, RES-loaded niosomes were produced using sonication. These carriers increased RES solubility and antioxidant activity [511].

#### 3.2.13. Resveratrol Clinical Trials

Clinical trials of RES have investigated its potential therapeutic activity in a number of diseases including cancer, obesity, neurological, cardiovascular, and infections. The emphasis in this review is on the role of RES as an anticancer. Although the clinical use of RES is highly limited by its stability and bioavailability issues, clinical trials showed that it was active either alone or in combination. This part of the review discusses the completed clinical and pharmacokinetics studies of RES from the published articles [512].

The safety of RES has been evaluated in healthy subjects, and it has been shown to be safe up to the doses of 5 g/d [513]. Colon cancer, breast cancer, and multiple myeloma are the most common cancers shown to respond positively to RES.

In one of the clinical trials conducted, the safety and efficacy of RES was evaluated in 40 healthy volunteers [513]. RES was found to be safe and to reduce the levels of IGF-1 and IGFBP-3, signaling molecules linked with several cancer types.

A phase I study was performed to assess the effect of low-dose RES (80 mg/d) and RES-containing freeze-dried grape powder (GP) (80 g/day which is equivalent to 450 g of fresh grapes) on colon cancer. After 14 days of treatment, an increase in the expression of myc and cyclin D1 was found in the colon cancer tissue. The GP was found to produce more pronounced effects compared with RES [514]. Additionally, GP significantly decreased CD133 with a mild effect on LGR5 (the downregulation of CD133 and LGR5 is linked with growth inhibition of colon cancer cells) in normal colonic mucosa [515].

Although RES has shown some efficacy in cancer patients, poor bioavailability limits its use. Therefore, efforts have been made to modify resveratrol for improved bioavailability and reduced toxicity [516].

In another study, micronized RES (SRT501) was used to enhance the absorption of RES across the gastrointestinal tract. SRT501 was used at 5 g/day, for two weeks, in patients with colorectal cancer and hepatic metastases. SRT501 was better tolerated and was shown to increase the mean plasma RES levels (3.6-fold) compared with the nonmicronized RES after a single dose [517].

In a randomized placebo-controlled clinical study in women with high breast cancer risk, the effects of RES on the expression of some cancer-related genes, such as CCND-2, p16, RASSF-1α, and cancer-promoting prostaglandin E2 (PGE2), were assessed [518]. The volunteers received two capsules per day, for 12 weeks, containing either placebo, 5 mg of *trans*-RES, or 50 mg of *trans*-RES. The PGE2 levels were found to be lowered.

Although these observations provide support for the potential effect of RES against breast cancer, they are based on a single clinical trial and further validation in a larger cohort of patients is needed.

### 3.3. Epigallocatechin-3-Gallate (EGCG)

#### 3.3.1. Formulations and Delivery Systems

To enhance its bioavailability and allow it to reach the highest levels in plasma, EGCG was taken after an overnight fasting period, together with 200 mg ascorbic acid and 1000 mg omega-3 fatty acids [519]. Moreover, taking EGCG on an empty stomach at least 30 min before breakfast [519], softening hard drinking waters [520] or adding sucrose (which enhances absorption in the digestive tract) may improve EGCG bioavailability [521]. The black pepper alkaloid piperine could serve as a potential dietary modulator of the bioavailability of EGCG by inhibiting its glucuronidation in the small intestines, as well as inhibiting gastric emptying and gastrointestinal transit, which may result in increased absorption [522]. Promising approaches to improving EGCG bioavailability, such as the encapsulation of EGCG in nanoparticles, improving intestinal absorption through polyphenol stabilization [523], the design of *O*-acyl derivatives of EGCG [524], the solid-phase synthesis of EGCG derivatives [525] or considering other delivery systems and routes of administration such as transdermal delivery [526] are being undertaken.

A colloidal vesicular system of EGCG for prevention and treatment of skin cancer was developed to overcome the problems associated with the chemical instability and low bioavailability of EGCG. EGCG was encapsulated in ultradeformable colloidal vesicular systems (penetration-enhancer-containing vesicles (PEVs), ethosomes, and transethosomes (TEs) for topical administration in skin cancer [527]. In addition to their reasonable skin deposition and preservation of the antioxidant properties of EGCG and physical stability, EGCG-loaded PEVs and TEs showed an inhibitory effect on epidermoid carcinoma in vitro and reduced tumor sizes in mice [527].

In another study, EGCG was encapsulated in solid lipid nanoparticles (EGCG-SLNs) to enhance its stability for anticancer activity. In vitro studies showed that the cytotoxicity of EGCG-SLNs was found to be 8.1 times higher against human breast cancer cells and 3.8 times higher against human prostate cancer cells, compared to the pure form of EGCG [528].

EGCG was also encapsulated with chitosan nanoparticles in order to improve bioavailability and intestinal absorption [529]. Heat treated β-lactoglobulin protected the antioxidant properties and improved stability and release of EGCG within the digestive tract [530,531]. Liposomes and gelatin have been useful in improving the stability and bioavailability of catechins in vivo and in vitro [532].

Stability enhancers of EGCG include antioxidants such as ascorbic acid [533]. However, ascorbic acid was found to destabilize EGCG in the presence of sucrose [534]. Other antioxidants include propyl gallate (Pgal) and butylated dihydroxytoluen (BHT) [535]. Additionally, amphiphilic compounds and other stabilizers such as glycerin and Transcutol P were shown to enhance its stability [535]. Na_2_EDTA (14 mM) was also reported to increase the stability of EGCG as it can scavenge metal ions which catalyze EGCG auto-oxidation [536].

Acetylation of EGCG can reduce the water solubility and enhance the stability and encapsulation of EGCG [537]. Moreover, esterification can increase the lipophilicity and bioavailability of EGCG [538].

Cationic polysaccharides, such as chitosan (CS) are suitable for EGCG oral delivery [539]. Intravenously administered delivery systems such as nanoparticles are passively targeted into tumors as a result of an enhanced permeability and retention effect in addition to targeted drug delivery [540].

Local delivery methods including topical, ocular, and intratumoral routes of administration avoid rapid renal clearance and significantly reduce the systemic side effects [533].

#### 3.3.2. Clinical and Epidemiological Studies

In this part, the focus is on the studies which reported significant association between green tea intake and cancer, as it seems to be more associated with reduced cancer risk, as compared to black tea.

A cohort study on 481,563 volunteers aged 51–71 years, and after up to eight years of follow-up, showed a statistically significant inverse relationship between consumption of hot tea and risk of pharyngeal cancer [541]. In a randomized placebo-controlled phase II clinical trial conducted on 59 patients to examine the effect of green tea extract on the oral mucosa leukoplakia (precancerous lesion of oral cancer), 37.9% of patients who received green tea treatment showed reduced size of oral lesions [542].

In another completed randomized placebo-controlled phase II trial to assess the potential of green tea extract to prevent oral cancer conducted on 42 patients who received oral amounts of 500, 750, or 1000 mg/m^2^ of green tea extract per day or placebo, 50% of the patients who completed the trial had a favorable response in a dose-dependent fashion [543].

The only epidemiologic study (a nested case-control study within the Shanghai Cohort Study) that examined specific tea catechins related to the risk of esophageal cancer with using validated urinary biomarkers for tea polyphenol uptake and metabolism, showed a decreased risk for both esophageal and gastric cancer with the presence of EGC in urine, with a stronger inverse association in nonsmokers or nondrinkers of alcohol or among those with lower serum levels of carotenes [544].

A meta-analysis that included 13 (five cohort and eight case-control) studies, reported an inverse association between the consumption of green tea and the risk of stomach cancer in case-control studies only but not in cohort studies [545].

A more recent pooled analysis of six cohort studies found a statistically significant, inverse association between green tea consumption and stomach cancer risk in women (primarily among female nonsmokers), but not in men [546].

A case-control study (nested within a prospective cohort of Chinese men in Shanghai, China) directly suggests a protective role of the tea catechin EGC on stomach cancer [544] as urinary levels of tea catechins were significantly associated with the reduced risk of stomach cancer.

A meta-analysis including 25 epidemiological studies in 11 countries [547], concluded an inverse association between green tea intake and colon cancer risk in four case-control studies.

Another prospective study (a cohort of 69,710 Chinese women aged 40 to 70 years, most of whom were lifelong nonsmokers or nondrinkers of alcoholic beverages) was conducted to evaluate the association between green tea consumption and colorectal cancer risk and reassessed two to three years later, in a follow-up survey. The study concluded that regular tea intake had significantly reduced risk of colorectal cancer [548].

Another prospective study provided evidence for the effect of tea catechins against the development of colon cancer. It examined the association between the urinary levels of EGC, 4′-*O*-methyl-epigallocatechin (4′-MeEGC) and EC, and their metabolites and the risk of developing colorectal cancer. Individuals with high levels of urinary catechins had a lower risk of colon cancer. However, there was no association between urinary green tea catechins or their metabolites and rectal cancer [549].

A randomized placebo-controlled phase II clinic trial supported a protective role of green tea polyphenols against two established risk factors for liver cancer, namely, aflatoxin and hepatitis B [550]. A population-based case-control study showed a statistically significant inverse association between the risk of pancreatic cancer with green tea consumption [551].

Given the short survival and rapid progression of pancreatic cancer, the available epidemiological data are insufficient to draw a conclusion whether green tea may protect against the development of pancreatic cancer or not.

In a meta-analysis including 22 studies, the risk of lung cancer was significantly decreased with green tea consumption. The decreased risk was confined to non-smokers and the association was slightly stronger for prospective cohort studies compared with retrospective case-control studies [552].

In a phase II randomized controlled tea-intervention trial to assess the effect of regular green tea intake on reducing DNA damage (through its antioxiative properities) among heavy smokers, a statistically significant decrease in DNA damage was reported. In the same study, no association was found in the black tea group [553].

In a pilot clinical study involving ten female patients (38–55 years old) with locally advanced noninflammatory breast cancer undergoing radiotherapy, EGCG capsules (400 mg) were orally administered three times daily, for two to eight weeks. EGCG was found to potentiate the efficacy of radiotherapy in breast cancer patients, and raise the potential of EGCG to be a therapeutic adjuvant in the treatment of metastatic breast cancer [554].

A meta-analysis, including seven epidemiological (two cohort, one nested case-control, and four case-control) studies to evaluate the effect of green tea on breast cancer, reported an inverse association between green tea and breast cancer risk only in the case-control data [555].

Two prospective cohorts [556,557], reported a statistically significant decrease in risk of breast cancer recurrence associated with green tea intake. Two studies based on prediagnostic biomarkers of tea intake and metabolism on the risk of breast cancer [558,559] found no association between urinary levels of any biomarkers measured and the risk of breast cancer [559].

While some case-control studies reported a statistically significant inverse relationship between green tea intake and prostate cancer risk [560], prospective cohort studies found no association [561]. Four interventional studies have been conducted to evaluate the effect of green tea intake on the change of risk biomarkers of prostate cancer [562,563,564,565]. Three were single arm open label phase II trials.

In the first trial which involved 42 patients with androgen-independent prostate cancer, one of the patients showed a 50% decrease in prostate-specific antigen (PSA) level, with this decrease sustained up to two months [562]. The second trial was to evaluate the efficacy and toxicity of standardized green tea extract on prostate cancer, 40% of the pateints who completed the therapy had delayed disease progression [563].

The third trial which was conducted to assess the effect of standardized green tea polyphenols during the interval between prostate biopsy and prostatectomy, showed that the supplementation (containing 1300 mg tea polyphenols or 800 mg green tea catechins) significantly reduced the levels of several cancer-related risk biomarkers [564].

According to the fourth trial, which was a randomized double-blind placebo-controlled phase II trial to assess the effect of green tea catechins on prostate cancer with high-grade prostate intraepithelial neoplasia, no clinical effect was observed [565]. Another study with a two-year follow-up showed a protective effect of green tea against the development of prostate cancer [566].

Five case-control studies [567,568] reported a statistically significant increased risk of bladder cancer associated with green tea or black tea consumption. In another cohort study with six years of follow-up, black tea intake was inversely associated with bladder cancer risk in a dose-dependent manner [569]. In a population-based case-control study [570], there was no association between tea consumption and bladder cancer risk.

Two hospital-based case-control studies found that green tea consumption was associated with a statistically significant decreased risk of leukemia [571,572].

In conclusion, a large number of studies have evaluated the association between tea consumption and risk of various cancers. However, the data obtained from these studies are sometimes conflicting or inconsistent.

The inconsistency and the varying results of tea-cancer associations might be due the relatively low levels of tea or tea polyphenols consumed in some human populations and the varying contents of tea catechins in different types of tea.

Other factors, such as the thermal effect of tea beverage, total fluid intake, and confounding effects of smoking and alcohol, may contribute to the inconsistent results.

Green tea seems to be more associated with reduced cancer risk in comparison with black tea, possibly due to the relatively high concentrations of catechins in green tea than in black tea.

Using validated biomarkers of the uptake and metabolism of tea polyphenols provides more reliable measurements and thus consistent results compared with relying on self-report.

### 3.4. Allicin

#### 3.4.1. Formulations and Delivery Systems

Although allicin is poorly stable and short-lived, it can easily pass through cell membranes owing to its hydrophobic nature. In cellular compartments, allicin reacts rapidly with free thiol groups [573].

The optimum activity of allinase enzyme, which converts alliin to allicin, is at pH 7.0 and becomes inactivated at pH values below 3.5 or with heating [574,575]. Thus, an enteric-coated formulation has been adopted to prevent stomach disintegration to many brands of garlic supplements and to protect allinase enzyme [576].

A microparticulate formulation, in which alliin and alliinase are separately encapsulated into microspheres, has been developed for pulmonary application [577].

In a study to determine the bioavailability of allicin in 23 types of garlic products in healthy individuals (six females and seven males) after 32 h of consumption, results showed allicin bioavailability of 26–109% for garlic powder capsules, 36–104% for enteric tablets, 80–111% for non-enteric tablets, 16% for boiled, 30% for roasted, 19% for pickled, and 66% for acid-minced garlic foods [576].

A strategy to improve the chemical instability of allicin was by chemical conjugation of the alliinase enzyme to a monoclonal antibody for a specific pancreatic cancer marker. This conjugate effectively induced apoptosis in MIA PaCa-2 cells [578].

Moreover, liposomes encapsulation improved the stability of allicin by protecting it from unfavorable conditions. This also diminished its characteristic unpleasant odor [579].

Allicin was loaded on locust bean gum nanoparticle (LBGAN). This system was shown to provide protection and stability, and it enhanced the pharmacological activity of allicin. Moreover, locust bean gum (LBG) which is a natural food additive has been reported to be effective against colon cancer [580].

More recently, several stabilized allicin derivatives were synthesized and tested for their activity on drug-sensitive (MCF-7) and multidrug-resistant (MCF-7/Dx) human breast cancer cells. Some of these derivatives were more effective than free allicin on inducing apoptosis [581].

#### 3.4.2. Clinical and Epidemiological Studies

Although allicin was shown to have a remarkable in vivo antitumor activity on various cancer types, this has not been followed up by a comparable number of human trials [582].

There is only one clinical trial recorded on clinicaltrials.gov on allicin application in cancer (follicular lymphoma NCT00455416), with no published results.

A double-blind randomized controlled trial involving patients with colorectal adenomas concluded a high dose of aged garlic extract was associated with a signicantly reduced risk of new colorectal adenomas [583].

In another randomized multi-interventional trial with 7.3 years of follow-up, administering 800 mg garlic extract plus 4 mg steam-distilled garlic oil daily [584], the prevalence of precancerous gastric lesions did not decrease, and gastric cancer incidence was not significantly affected [585].

Clinical trial data of various forms of garlic are inconsistent because of the significant difference in the bioavailability of the constituents in raw garlic and the specific garlic supplement formulations [586].

In aged garlic extract (AGE), garlic is aged for up to 20 months, a process that converts the odorous, harsh and irritating compounds of garlic into stable and safe sulfur compounds [587].

Aged garlic extract was shown to reduce the incidence and proliferation of colorectal cancer [588]. Administering aged garlic in patients with advanced cancer of the digestive system improved natural killer (NK) cell activity but did not improve the quality of life (QoL) [589]. Moreover, large doses of allitridum and microdoses of selenium were shown to prevent gastric cancer, especially in men [590].

Although some association between increased intake of onions and garlic and decreased risk of certain cancers was supported by epidemiological studies, the data are limited and sometimes conflicting.

The major epidemiological evidence suggests protective effects of garlic and/or onions against gastrointestinal cancers. These observations were based on recent systematic reviews and meta analyses [591].

A meta-analysis involving 19 case-control and two cohort studies, showed a reduced risk of gastric cancer with an increment of 20 g/day of total *Allium* vegetables including garlic, onion, leeks, Chinese chives, scallions, and garlic stalks [592].

Another meta-analysis involving 14 case-control studies that investigated the effect of *Allium* vegetables on stomach cancer and five case-control studies that investigated the effect of garlic on stomach cancer, concluded a potential cancer preventive effect of *Allium* vegetables on stomach cancer [591].

Data of epidemiological studies on colorectal cancer are conflicting. Some meta-analysis studies [593] showed no decrease in the risk of colorectal cancer with increased *Allium* consumption. Other case-control studies (1037 cases and 2020 controls) [594] found that both onions and garlic were protective against cancers of the large bowel.

Another study conducted to evaluate the effect of consuming various foods including raw garlic/onion on the development of esophageal squamous cell carcinoma involving 343 patients with esophageal squamous cell carcinoma and 755 cancer-free control, concluded that the consumption of raw garlic/onion at least once per week significantly protects against esophageal squamous cell carcinoma [595].

In a population-based case-control study [596], individuals in the highest of three intake categories of total *Allium* vegetables had a 53% decreased incidence of prostate cancer compared to those with the lowest intake. Both garlic and scallion alone were also associated with decreased incidence, while leeks, Chinese chives, and onions were not.

However, in another study, garlic supplement use was not associated with a decreased risk of prostate cancer [597].

The associations between onions or garlic and cancers of the oral cavity/pharynx, larynx, renal cells, breast, ovary, and endometrium were investigated [594,598]. It was reported that there is significant inverse associations between onion intake of seven or more times per week and oral cavity/pharyngeal cancers, laryngeal cancer and ovarian cancer. Moreover, significantly decreased odds of laryngeal and ovarian cancers were associated with one-to-seven servings of onions/week. Servings of more than twice a week was associated with decreased risk of endometrial cancer. Furthermore, high garlic intake resulted in decreased odds of oral cavity/pharyngeal, laryngeal, ovarian, renal cell and endometrial cancers. The associations observed between onions or garlic and breast cancer, or for onions and renal cell carcinoma were significant. Another case study where controls were randomly selected from a list of residents found that high intake of onions, but not garlic, was associated with decreased risk of lung cancer [599], with stronger association with decreased risk of squamous cell carcinoma than with adenocarcinoma. Vitamins and lifestyle (VITAL) cohort studies showed that high intake of garlic supplements for over 10 years was associated with 45% decreased odds of hematological malignancies [600].

### 3.5. Emodin

#### 3.5.1. Formulations and Delivery Systems

A variety of approaches to improve the solubility of emodin have been evaluated. Liposomal emodin improved the physical stability and provided an appropriate circulation time in the blood. Liposomal-emodin was conjugated with d-α-tocopheryl polyethylene glycol 1000 succinate (TPGS) to improve the encapsulation efficiency and stability of emodin egg phosphatidylcholine/cholesterol liposomes. This system was compared with methoxypolyethyleneglycol 2000-derivatized distearoyl-phosphatidylethanolamine (mPEG2000-DSPE) liposomal emodin and showed more improved cytotoxicity of emodin on leukemia cells and longer circulation time in the blood. Moreover, higher AUC of emodin in the lungs and kidneys was achieved with TPGS liposomes compared to mPEG2000-DSPE liposomes, with the comparable elevated amount of emodin in the heart for both liposomes [601].

Silk-fibroin-coated 1,2-dimiristoyl-*sn*-glycero-3-phosphocholine/Tween20 liposomes of emodin were significantly more effective on breast cancer cells than nontargeted liposomal emodin [602,603].

Emodin was also formulated using poloxamer-based (poloxamer 407, poloxamer 188, and PEG400) thermoreversible gel for topical delivery. The solubility of emodin was enhanced, at least by 100-fold, compared to 10% ethanol or water. This formulation enhanced cellular uptake by the human dermal fibroblast cell line and DLD-1 colon cancer cell line [604].

In addition, emodin loaded onto solid lipid nanoparticles (E-SLNs) was prepared to improve its anticancer efficacy. Compared to free emodin, this system showed sustained release and significantly higher cytotoxicity against human breast cancer MCF-7 cells and human hepatoma HepG2 cells [605].

E-SLNs prepared using poloxamer 188 and Tween 80 as surfactants exhibited enhanced physical stability and sustained profile. This makes this carrier a promising carrier for oral drug delivery. Moreover, E-SLNs significantly enhanced the in vitro cytotoxicity against human breast cancer cell line MCF-7 and MDA-MB-231 cells [606].

Inclusion of emodin in hydroxypropyl-β-cyclodextrin (emodin/HP-β-CD) remarkably enhanced the water solubility of the compound, as well as its in vitro cytotoxicity compared with emodin alone [607]. In another study, mesoporous silica SBA-15 was used as a vehicle for the transport of emodin to protect it from the stomach acidic conditions and from photodecomposition. In vitro studies to evlaute this system on the tumor cell lines melanoma A375, B16, and B16F10 showed tumor antiproliferative and apoptotic effect [608].

#### 3.5.2. Emodin Clinical Studies

Searching the https://clinicaltrials.gov/ website for clinical studies for the anticancer activity of emodin returned only one study on the effect of emodin on breast cancer, but with unknown status (ClinicalTrials.gov Identifier: NCT01287468).

### 3.6. Thymoquinone

#### 3.6.1. Formulations and Delivery Systems

Thymoquinone has a good safety profile and exhibits anticancer activity at very small doses, less than 10 mg/kg [609]. The clinical limitation of thymoquinone in humans is due to its hydrophobicity which is responsible for the poor formulation characteristics and thus poor bioavailability, and poor membrane penetration capacity [610].

Chemical derivatives have been developed to improve the bioavailability of thymoquinone. Thymoquinone-4-α-linolenoylhydrazone and thymoquinone-4-palmitoylhydrazone are examples for derivatives of thymoquinone which were found to inhibit cell proliferation with improved bioavailability [154]. Caryophyllyl, germacryl, and fatty acid conjugate analogues of thymoquinone showed a significantly more potent activity against sensitive and resistant MCF-7 breast cancer cell lines compared to thymoquinone [611,612].

Moreover, nano-formulations were very effective in enhancing the bioavailability of thymoquinone. Thymoquinone-loaded liposomes, in which the liposomes were modified with Triton X-100 (XLP), were found to improve the stability and bioavailability, and maintain the anticancer activity of thymoquinone [610]. Additionally, nanoparticulate formulation based on PLGA and polyethylene glycol (PEG)-5000 was found to enhance the antiproliferative, anti-inflammatory, and chemosensitizing effects of thymoquinone [613].

Thymoquinone formulated with double mesoporous cor-shell silica spheres was more effective in inducing apoptosis compared with the free thymoquinone, due to the slow release from the mesoporous structure [614].

Thymoquinone-encapsulated nanoparticles, using hydrophilic polymers such as polyvinylpyrrolidone and polyethyleneglycol, were effective in enhancing thymoquinone’s solubility, reducing its thermal and light sensitivity, and enhancing systemic bioavailability. This system can induce apoptosis of breast cancer cells and reduce migration [615].

Thymoquinone-loaded into myristic acid chitosan nanogel [616] was found to have more effective anticancer activity on human breast adenocarcinoma cells MCF-7 than thymoquinone alone.

Chitosan NPs prepared by ionic gelation displayed burst release followed by sustained slow release with high targeting to the brain after intranasal administration [617]. Solid lipid nanoparticles (SLNs) prepared by ultrasonication showed an initial rapid release phase followed by a slower sustained release phase, with a two-fold increase in the bioavailability of thymoquinone after oral administration in rats [618]. SLNs prepared by solvent injection with controlled time-dependent release of thymoquinone showed a five-fold increase in the bioavailability of thymoquinone after oral administration in rats, higher concentrations in major organs and lower liver damage [619].

#### 3.6.2. Current Clinical Trials

Although there is a large number of in vitro and in vivo studies that revealed the potential of thymoquinone as an anticancer agent, there is no clinical application yet. There are no clinical trials for thymoquinone registered on (https://clinicaltrials.gov/).

### 3.7. Genistein

#### 3.7.1. Formulations and Delivery Systems

Genistein has high clinical development value as an anticancer drug because it can be orally administered, and has less toxicity and side effects than other anticancer drugs. The clinical applications of genistein have been restricted because of its low bioavailability, biological estrogenic activity and detrimental effects on thyroid function. The bioavailability of genistein can be enhanced by fermentation, use of micromicelles and modification of its chemical structure. One of the strategies to enhance the bioavailaibilty of genistein is the fermentation of soy products where genistein mainly exists. Fermentation transforms most isoflavones into isoflavone glycoside ligands which have high bioavailability and accelerated absorption [620].

Shen and co-workers developed micromicelles of genistein which significantly improved its water solubility, membrane permeability and oral bioavailability [621].

Wang and co-workers added phosphate groups into genistein to form genistein-*O*-phosphatidic acid. The solubility of genistein-*O*-phosphatidic acid in water, permeability of small intestine and thus genistein plasma concentration were greatly enhanced [622]. Moreover, stability of genistein against metabolism was enhanced by methylating the free hydroxymethyl group of genistein [623].

#### 3.7.2. Genistein Clinical Studies

A meta-analysis on peri-menopausal women indicated that increased intake of soy food reduced the incidence of breast cancer [624]. Another meta-analysis involving 9000 breast cancer cases suggested that increasing the genistein dose would reduce the risk of breast cancer [625]. However, low doses of genistein inhibit the proliferation of breast cancer cells, while high doses of genistein promote the proliferation of breast cancer cells in in vitro settings [626].

Two meta-analysis studies showed that genistein was able to postpone diffusion and invasion of prostate cancer and enhance intercellular adhesion [627,628].

In a clinical study involving 140 patients with an early phase breast cancer where the group received a soy-supplemented diet (25.8 g/day) for 30 days, high genistein concentration in plasma was associated with increased proliferation of breast cancer cells [629].

On the other hand, a phase I clinical trial study including patients with prostate cancer treated with different doses (including 2, 4, and 8 mg/kg/day) of genistein showed no obvious adverse reactions [630]. Another study on patients with prostate cancer, who received a daily dose of genistein at 300 or 600 mg, for 84 days, indicated no obvious adverse reactions [631]. This study was followed by a phase II clinical trial where patients with prostate cancer received genistein for four weeks. The expression of MMP-2 in the genistein-treated group was 76% less than that in the control group [632]. Moreover, a randomized placebo-controlled double-blind phase II clinical trial including prostate cancer patients received genistein before radical prostatectomy indictaed that the level of the prostate cancer biomarker prostate-specific antigen (PSA) in blood was reduced compared to the control [633]. 

In a phase I/II pilot clinical trial (ClinicalTrials.gov Identifier: NCT01985763) to evaluate the efficacy of genistein in the treatment of metastatic colorectal cancer alone or combined with 5-fluorouracil and platinum compounds, it has been revealed that genistein may inhibit Wnt signaling, a pathway activated in the majority of colorectal cancers, and augment growth inhibition when combined with 5-fluorouracil and platinum compounds.

A phase II trial (ClinicalTrials.gov Identifier: NCT00376948) on patients with locally advanced or metastatic pancreatic cancer, showed that genistein in combination with gemcitabine and erlotinib can help kill more tumor cells by making tumor cells more sensitive to the drugs.

A randomized phase II trial (ClinicalTrials.gov Identifier: NCT01325311) on patients with early stage prostate cancer demonstrated that cholecalciferol (200,000 IU) and genistein (as G-2535 which provides 600 mg of genistein) may slow the growth of cancer cells and may be an effective treatment for prostate cancer.

### 3.8. Parthenolide (PTL)

#### 3.8.1. Formulations and Delivery Systems

Various delivery systems were developped to enhance the aqueous solubility and biovailablity of PTL. These systems include micelles formed from the copolymers of poly (styrene-alt-maleic anhydride)-b-poly (styrene) (e.g., PSMA_100_-b-PS_258_). PTL-loaded PSMA-b-PS micelles exhibited a dose-dependent cytotoxicity towards acute myeloid leukemia AML cells and were capable of reducing cell viability by 75% at 10 μM PTL [634].

Carboxyl-functionalized nanographene (fGn) delivery system was used to overcome the extreme hydrophobicity of PTL. Gn offers a facile modifiable and customizable system for drug loading and delivery [635]. This system was compared with the water-soluble analogue of PTL, dimethylamino parthenolide (DMAPT) for their anticancer efficacy [636]. The fGn system is postulated to enhance the activity of PTL by assisting in the drug-internalization process, enhancing the water-solubility of the drug and improving cellular drug uptake Delivery by fGn was found to enhance the anticancer/apoptotic profile of PTL when delivered to the human pancreatic cancer cell line, Panc-1, while DMAPT was not [635].

Additionally, PLT-combined administration with ginsenoside compound K (CK), using the tumor-targeting carriers tLyp1 liposomes, was found to enhance tissue penetration and selectively target neuropilin-1 on the surface of lung cancer cells, thus facilitating the delivery of PTL and CK to the tumor. In vivo studies showed that PTL/CK tLyp-1 liposomes produced a greater antitumor effect against lung cancer than combined administration of these compounds, with minimal toxicity [637].

‘Stealth liposomes’, prepared from egg phosphatitylcholine (EPC), cholesterol, and polyethylene glycol-distearoyl phosphosphatidyl ethanolamine (PEG2000-DSPE), were used to encapsulate PTL individually and in combination with drugs such as vinorelbine for targeting breast cancer cells and CSCs. PTL-loaded liposomes efficiently killed the CSCs, and the PTL/vinorelbine liposomes completely inhibited tumor growth in vivo [638].

PTL was also incorporated into liposome systems composed of phosphatidylcholine and 1,2-distearoyl-*sn*-glycero-3-phosphoethanolamine poly(ethylene glycol) 2000 (DSPE-PEG 2000) in combination with three natural products in liposome systems, namely, betulinic acid, honokiol and ginsenoside Rh2 for lung cancer treatment. This cocktail liposome system was shown to provide a more efficient and safer treatment for lung cancer [639].

#### 3.8.2. Clinical Studies

There are no clinical studies conducted to evaluate the anticancer activity of parthenolide available on the https://clinicaltrials.gov/ website.

### 3.9. Luteolin

#### Formulations and Delivery Systems

Poor water solubility, bioavailabilty, and extensive excretion of luteolin have significantly limited its clinical use [640]. Searching clinicaltrials.gov returned only one clinical study to evaluate the effect of luteolin (nano-luteolin) on tongue carcinoma which has passed its completion date with unkown status.

To improve the oral bioavailability of luteolin, it was encapsulated with zein protein and sodium caseinate. These nanoparticles of luteolin exhibited enhanced cytotoxicity against colon cancer cells (SW480) and induced apoptosis [641].

Vitamin E d-α-tocopherol acid polyethylene glycol 1000 succinate (TPGS)-coated liposomes showed enhanced cellular uptake and apoptosis, and targeted delivery of luteolin in vivo and in vitro. In addition to its excellent emulsifying and solubilizing properties [642], TPGS was found to inhibit *p*-gp-mediated multidrug resistance, and enhance the absorption and cytotoxicity of drugs [643].

Solid lipid nanoparticles (SLNs) have been shown to improve the bioavailability of luteolin [644]. Moreover, the solubility of luteolin was improved approximately 2.5 times compared to free luteolin when complexed with phospholipids [645].

Using a process, named supercritical assisted injection in a liquid antisolvent (SAILA), luteolin was entrapped in zein microparticles. This system showed faster dissolution rate with a consequent increase in bioavailability while preserving the antioxidant activity of luteolin [646].

The solubility and oral bioavailability of luteolin was also enhanced by applying the super-saturable self-nanoemulsifying drug delivery system (S-SNEDDS). This system is consisted of caprylic/capric triglyceride, polyoxyl 35 hydrogenated castor oil and polyethylene glycol 400 [647].

Additionally, luteolin formulated as PLA-PEG showed a higher inhibitory effect on tumor growth compared with the free luteolin when tested on athymic nude mice and Tu212 cells (laryngeal cancer cells) using a dose of 3.3 mg/kg [648].

### 3.10. Quercetin

#### 3.10.1. Formulations and Delivery Systems

Quercetin has low bioavailability due to its low water solubility and unstability in physiological conditions [649]. Nano-formulations including microemulsion, nanoparticles, liposomes and solid lipid nanoparticles have been developed for quercetin to enhance its bioavailability [650].

Encapsulation of quercetin into biodegradable monomethoxy poly(ethylene glycol)-poly(ε-caprolactone) (MPEG-PCL) micelles resulted in complete dispersion of quercetin in water and presented a promising potential in the treatment of ovarian cancer [651].

Quercetin was also formulated as PEGylated liposomal quercetin (Lipo-Que). In vitro studies revealed that Lipo-Que induced apoptosis and suppressed cell proliferation in both cisplatin-resistant (A2780cp) and cisplatin-sensitive (A2780s) human ovarian cancer models [652].

#### 3.10.2. Quercetin Clinical Studies

A phase I clinical trial to evaluate the safety profile of quercetin concluded that quercetin can be safely administered by intravenous bolus injection. Additionally, quercetin was shown to inhibit lymphocyte tyrosine kinase [653].

There is only one completed clinical study to evaluate the effect of quercetin related to the prevention and treatment of cancer (chemotherapy-induced oral mucositis) but with no results posted (https://clinicaltrials.gov/).

### 3.11. Paclitaxel (Taxol)

Paclitaxel is sold under the brand name Taxol^®^ since 1993. It is practically insoluble in aqueous medium, and hence it is administered in a solution containing alcohol and polyoxyethylated castor oil [654].

The development of new drug delivery systems allowed paclitaxel to overcome its multidrug resistance, poor aqueous solubility, clinical neurotoxicity, and neutropenia [654]. 

The first injectable paclitaxel lecithin/cholesterol liposome, Lipusu^®^, has been used in the treatment of ovarian, breast, non-SCLC, gastric, and head and neck cancers [655]. Lipusu^®^ significantly lowered the toxicity of paclitaxel [655,656]. Abraxane^®^ is another injectable nanoparticle albumin-bound delivery system for paclitaxel that improved the solubility of paclitaxel, with improvement in neuropathy side effects after therapy discontinuation. It was approved in 2005 by FDA and in 2012 by European Medicines Agency (EMA) [657].

Moreover, semisynthetically developed paclitaxel mimics, with a simplified structure, fewer side effects and improved pharmaceutical properties have been developed. This allowed the discovery of docetaxel (on the market since 1995) [658]. Another example for paclitaxel mimics is cabazitaxel (Jevtana^®^) which was approved by the FDA for the treatment hormone-refractory metastatic prostate cancer and docetaxel- or paclitaxel-resistant tumors [659].

Paclitaxel is already tested in combination with other anticancer drugs such as nilotinib (ClinicalTrials.gov Identifier: NCT02379416), carboplatin and megesterol acetate (ClinicalTrials.gov Identifier: NCT00584857), and bavituximab (ClinicalTrials.gov Identifier: NCT01288261).

### 3.12. Vincristine

#### 3.12.1. Formulations and Delivery Systems

The clinical use of vincristine as an anticancer has been approved by the Food and Drug Administration (FDA) in 1963 [660]. It has been used in adult and paediatric chemotherapy mainly against acute lymphoblastic leukemia (ALL). Incorporating vincristine in the treatment regimen was shown to increase the survival rate to 80% [661].

The major obstacles associated with the use of vincristine are its low natural occurrence and consequently its high cost of extraction, and its side effects such as peripheral neuropathy [662]. Encapsulation of vincristine into liposomes mitigates some of these factors by increasing the circulation time, optimizing delivery to target tissues and allowing dose increasing without toxicity [663].

In 2012, sphingomyelin/cholesterol (SM/Chol) liposomal vincristine (Marqibo^®^) was approved by the FDA to treat adults with relapsed ALL. SM/Chol liposomal vincristine exhibits a longer circulation time, a reduced leakage rate and an enhanced antitumor effect compared to PEGylated liposomal vincristine [664].

Co-encapsulation of vincristine and quercetin in lipid-polymeric nanocarriers (LPNs) exhibited a synergistic effect and presented a potential approach to overcome chemo-resistant lymphoma [665]. Vincristine sulfate was incororporated into cetyl palmitate solid lipid nanoparticles using dextran sodium sulfate. This system exhibited comparable cytotoxic effects to the free drug against MDA-MB-231 cells and enhanced the half-life and concentration in plasma and brain tissue [666].

Folic acid/peptide/PEG PLGA composite particles are another material system used to incorporate vincristine. This system was able to enhance drug uptake in MCF-7 cells [667]. Additionally, encapsulating vincristine in self-assembled dextran sulphate–PLGA hybrid nanoparticles was able to overcome multidrug-resistant tumors [668].

Composite core-shell particles (with a PLGA core, a hydrophilic PEG shell, phosphatidylserine electrostatic complex, and an amphiphilic lipid monolayer on the core surface) exhibited sustained-release characteristics, and a greater uptake efficiency and toxicity to MCF-7/Adr cells [669].

Different formulas of PLGA, PEG, dextran sulphate, oleic acid, liposomes, chitosan (PLGA loaded collagen–chitosan complex film, poly(butylcyanoacrylate) (PBCA) nanoparticles modified with plironic F-127, transfersome (vincristine loaded Transfersome), and niosomal vincristine were prepared [670].

#### 3.12.2. Vincristine Clinical Studies

Various clinical trials have been conducted to establish the safety, efficacy, and pharmacokinetics of Marqibo in ALL and Philadelphia chromosome-negative ALL (ClinicalTrials.gov Identifier: NCT00495079), and malignant melanoma and hepatic dysfunction (ClinicalTrials.gov Identifier: NCT00145041). In general, Marqibo was able to improve the therapeutic index of vincristine and exhibit complete remission in relapsed ALL (ClinicalTrials.gov Identifier: NCT00495079).

Vincristine generally exhibits better efficacy when administered in combination with other antitumor agents. This also helps decrease toxicity and drug resistance. Various completed clinical studies (with posted results) have been conducted to evaluate the chemoprevention efficacy of vincristine in combination with other anticancer drugs.

### 3.13. Bromelain

#### 3.13.1. Formulations and Delivery Systems

Although numerous studies have been conducted on bromelain, there is limited literature that reports the anticancer activity of the complex. Orally administered bromelain is well absorbed by the gut without losing its biological properties [671].

Nanostructures of bromelain were developed using inorganic compounds such as mesoporous silica nanoparticles (MSNs) [672]. In general, MSNs developed with bromelain can enhance diffusion to tumor extracellular matrix [673]. Gold nanocarriers which allows surface functionalization, is another strategy [674,675]. Synthetic polymers used in the nanoparticle design of bromelain include poly acrylic acid (PAA) [676] and PLGA [677].

Encapsulation of bromelain exhibited an enhanced anticancer ability in skin carcinogenesis in mice models [677]. Bromelain encapsulated with hyaluronic acid (HA) grafted PLGA copolymer was delivered efficiently to various cancer cells, with a higher cytotoxicity. Intraperitoneal and intravenous administration of encapsulated bromelain showed that NPs were efficient in suppressing the tumor growth in animal models of in Ehrlich’s Ascites Carcinoma [678]. Chitosan derivatives such as lactobionic acid-modified chitosan (CLA) were found to be efficient tumor-targeting polymers [679].

Additionally, niosomal formulations were developed to incorporate the protease enzymes papain and bromelain. Bromelain-loaded elastic niosomes exhibited superior elastic property and entrapment efficiency compared to non-elastic niosomes [680]. Bomelain-surface functionalized lipid core nanocapsules (LCNs) exhibited improved antiproliferative effect against human breast cancer cells (MCF-7) [681].

#### 3.13.2. Bromelain Clinical Studies

No controlled clinical studies were conducted on bromelain as an anticancer. Early anecdotal clinical studies of bromelain provide suggestive evidence of its effect against some cancers such as breast and ovarian cancers [682,683]. Some clinical studies of bromelain have been completed but without results posted on clinicaltrials.gov.

### 3.14. Boswellic Acids (BAs)

#### 3.14.1. Formulations and Delivery Systems

BAs are marketed as tablets, capsules, and ointments. Although there is no standard dosage regimen for BAs, the oral dose of BAs extract, at 300–350 mg, three times a day, gives effective plasma concentration [684].

Several approaches have been used to enhance the bioavailability and brain levels of BAs. Administering BAs with a standardized meal [685] or with anionic drugs [686] was found to enhance its bioavalaibility.

Additionally, different formulations such as lecithin delivery form (PhytosomeR), nanoparticle delivery systems including liposomes, emulsions, solid lipid nanoparticles, nanostructured lipid carriers, micelles and poly (lactic-co-glycolic acid) nanoparticles, and synthetic derivatization have been used to overcome the bioavailability limitations of BAs [687,688,689]. The lecithin delivery system was found to improve absorption and tissue penetration of BAs in a single-dose as reported by a randomized open-label study [690].

Encapsulation of BAs with β-CD and hydroxypropyl- beta-cyclodextrin (HPβ-CD) improved their solubility and oral bioavailability profile [691]. Further, micellar solubilized delivery of *Boswellia* extract led to a remarkable increase in the AUC and C_max_ of all BAs in plasma [692].

Self-nanoemulsifying system (SNES) was used to enhance the bioavailability of BAs including 11-keto-β-boswellic acid (KBA) and acetyl-11-keto-β-boswellic acid (AKBA). The water solubility and oral bioavailability of KBA and AKBA were increased by two-to-three-fold [693].

AKBA intestinal absorption was enhanced from total BAs fraction using cyclodextrin (CD) and poloxamer solid dispersion (PXM SDs) formulations. HP-β-CD and PXM 407 were shown to effectively enhance intestinal absorption through improved solubility [694].

Nanocarriers used in delivering and targeting BAs include the soy lecithin formulation of standardized *B. serrata* gum resin extract (CasperomeTM) [687]. After oral administration, CasperomeTM significantly increased plasma levels of KBA and βBA compared with the non-formulated extract. It also remarkably increased the concentrations of KBA and AKBA (35-fold), as well as βBA (three-fold) in the brain at a similar dose, and achieved 17-fold higher BAs levels in poorly vascularized organs such as the eye [687].

BAs nanoparticles containing 10 and 20 mg/kg BAs, investigated in animal models with prostate cancer, exhibited complete remission [695]. PLGA-based nanoparticle formulation of AKBA (AKBA-NPs) was found to enhance the oral bioavailability of AKBA [696].

#### 3.14.2. Boswellic Acids Clinical Studies

Boswellic acids (BAs), a series of pentacyclic triterpene molecules, were implicated in different phases of human clinical trials. As evidenced by a prospective randomized placebo-controlled double-blind pilot trial on 44 patients with primary or secondary malignant cerebral tumors randomly assigned to radiotherapy plus either *Boswellia serrata* (BS) 4200 mg/day or placebo, treatment with BS has been shown to decrease cerebral edema in patients irradiated for brain tumors by at least 75% [697]. Another study [698] conducted to investigate the use of the BS preparation H15 in 12 patients with cerebral edema, showed a clinical or radiological response in 8 of 12 patients.

The only completed clinical trial for cancer (but with no results posted) was a phase II, multicenter, self-controlled clinical trial to test the safety and efficacy of OPERA capsules (dietary supplement where α-lipoic acid, *Boswellia Serrata*, methylsulfonylmethane, and bromelain are combined in a single capsule) for treating breast cancer (ClinicalTrials.gov Identifier: NCT04161833). Table 2 summarizes the commercially approved target delivery systems and their active natural products.

## 4. Toxicity and Safety of Nanoparticles

Owing to their features of low toxicity and excellent physiochemical properties, nanoparticles (NPs) especially iron oxide nanoparticles have become a powerful platform in many aspects of biomedicine [702]. However, their widespread biomedical applications posed serious concerns about their safety. This part discusses the toxicity and safety considerations of nanoparticles (NPs), including metallic and non-metallic NPs.

### 4.1. Iron Oxide NPs

Iron oxide-based materials such as magnetite (Fe_3_O_4_) and maghemite (γ-Fe_2_O_3_) are considered the most suitable materials for synthesis of magnetic nanoparticles (MNPs) [703]. Maghemite is more preferred for the MNP cores because it is less likely to cause health hazards and as iron (III) ions are already found in the human body [704].

It is thought that the toxic effects of iron oxide NPs are due to excessive production of reactive oxygen species (ROS). These generated ROS further elicit DNA damage and lipid peroxidation [705].

Nanoparticle size, surface charge, shape, and stability are major contributing factors which determine the interaction of the iron oxide nanoparticles with proteins, and thus their fate and biodistribution inside the body [705].

Depending on the size, iron oxide NPs can be distributed to various organs, tissues, and cells. For instance, iron oxides NPs smaller than 10 nm are usually rapidly removed through renal clearance, whereas iron oxides NPs larger than 200 nm are sequestered by the spleen by filtration [706].

The iron oxide NPs are mainly distributed to the liver and to a lesser extent to spleen and in bone marrow [707]. Iron oxide NPs were also found to be able to cross the blood-brain barrier after inhalation [708].

The major biosafety issue related to iron oxide NPs is its interference with the iron released by physiological iron metabolism. Accumulation of intracellular iron can oxidize and damage the protein and nucleic acid components of cells [705], induce cell lysis, and disturb blood coagulation [709].

To enhance the stability of iron-based NPs, coating with other materials such as polymers and noble materials (gold and silver) can provide them with better stability, sustainability and mechanical strength [702].

Derivatives such as dextran-coated iron oxide NPs (100–150 nm, 0.1 mg/mL) [710], chitosan-coated iron oxide NPs (13.8 nm, 123.52 μg/mL) [711], and 1-hydroxy-ethylidene-1,1-bisphosphonic acid-coated iron oxide NPs (20 nm, 0.1 mg/mL) [712] were reported to decrease cell viability in in vitro models.

The potential of toxicity of iron oxide NPs greatly depends on the different combinations and functionalization used. In vivo and in vitro studies on the safety of uncoated iron oxide NPs displayed cytotoxicity in erythrocytes [713] and neurobehavioral toxicity at 10 ppm [714]. However, another study did not indicate any inherent toxicity for uncoated iron oxide NPs [702].

High exposure to charged NPs, was found to lead to charge-dependent fetal loss in an animal model [715]. PLGA-based iron oxide NPs exhibited concentration dependent cytotoxicity in BEL7402 cancer cells [716]. The increase in the surface charge of the NPs due to the chitosan coating enhanced the intracellular uptake of particles and thus increased their cytotoxic activity [717].

On the other hand, other combinations and formulas of iron oxide NPs including starch-iron oxide NPs and dextran-iron oxide NPs [718], dimercaptosuccinic acid-coated superparamagnetic iron oxide nanoparticles [719], gold-coated iron oxide NPs [720], amino acid-coated iron oxide NPs [721], PEGylated iron oxide NPs [722], multifunctional iron oxide NPs (with anti-CD47 antibody) [723], and manganese-based iron oxide NPs [724] did not show any potential toxicity in in vitro or in vivo models. However, polyethyleneimine (PEI)-coated iron oxide NPs exhibited dose-dependent lethal toxicity in BALB/c mice model [722].

Interestingly, *n*-octyltriethoxysilane-coated iron oxide NPs significantly decreased the cytotoxic effects [725]. Moreover, carbon-coated iron oxide NPs were reported to significantly reduce neurobehavioral toxicities compared to the bare MNPs [726].

### 4.2. Aluminum Oxide NPs

Aluminum oxide NPs were reported to alter the cell viability, mitochondrial function and tight junction protein expression of the blood brain barrier (BBB), and to increase oxidative stress [727]. However, some studies showed no significant toxic effects on the viability of mammalian cells (at concentrations of 10, 50, 100, 200, and 400 μg/mL) [728], and a dose-dependent cytotoxicity on human mesenchymal stem cells [729]. Moreover, dose-dependent genotoxic properties [730] have been associated with the use of aluminum oxide NPs.

### 4.3. Gold NPs

The Gold NPs are considered to be relatively safe [731]. A study to evaluate the cytotoxic effects of several gold NPs (4, 12, and 18 nm) with different capping agents against leukemia cell line [732], suggested that spherical gold NPs are not cytotoxic. However, some other reports indicated that cytotoxicity depends on the dose, side chain (cationic) and the stabilizer used [733,734]. It was also found to depend on the type of toxicity assay, cell line, and physical/chemical properties [735].

### 4.4. Copper Oxide NPs

Copper oxide NPs (50 nm) have been reported to have genotoxic and cytotoxic effects in addition to the ability to disturb cell membrane integrity and induce oxidative stress [736].

### 4.5. Silver NPs

In vivo study revealed that silver NPs have been detected in various organs including lungs, spleen, kidneys, liver, and brain after inhalation or subcutaneous injection [737]. Silver NPs were shown to have significant toxicity, induced by increased generation of ROS [738].

The cytotoxicity of silver NPs depends on the type of coating. For instance, using human lung cancer cell line, polyvinyl-pyrrolidone-coated silver NPs (6–20 nm) showed a dose-dependent cytotoxicity and cellular DNA adduct formation [739]. Other reports supported that peptide-coated silver NPs (20 nm) are more cytotoxic compared to citrate-coated silver NPs of the same size [740].

### 4.6. Zinc Oxide

The most common toxic effects of zinc-based nanomaterials were cell membrane damage and increased oxidative stress when tested on various mammalian cell lines [741,742]. Moreover, DNA damage, alteration in mitochondrial activity, almost complete cell death in the cell cultures [743,744] and genotoxicity [745] were reported after exposure to zinc oxide NPs (20 nm).

### 4.7. Titanium Oxide

Titanium oxide NPs displayed various toxic effects in experimental animals, including DNA damage and inflammation [746,747]. They were also reported to affect immune system, liver, kidney, spleen, myocardium, glucose, and lipids homeostasis in experimental animals [748,749].

### 4.8. Carbon-Based Nanomaterials

Carbon-based nanomaterials such carbon nanotubes have been reported as cytotoxic agents [731,750]. These effects vary with the size, method of preparation, and presence of trace metals in NPs [731].

### 4.9. Silica

NPs of silica induce the generation of ROS and subsequent oxidative stress, inflammatory biomarkers such as IL-1, IL-6, IL-8, TNF-α, and mitochondrial damage [731,751,752]. Silica-based NPs (70 nm) at 30 mg/kg were shown to alter biochemical parameters along with hepatotoxic effects [753].

### 4.10. NPs of Polymeric Materials

Polymeric-based NPs such as Poly-(d,l-lactide-co-glycolide)-based nanosystems did not show any cytotoxic, immunogenic or inflammatory effects. However, the surface coating of these NPs was found to induce the toxicity of polymeric NPs towards macrophages [754].

In addition to biocompatibility, NPs derived from certain bioceramic substances such as hydroxyapatite (calcium phosphate-based biomaterials) did not show any bio-accumulative toxicity in rabbits after intravenous injection [755,756]. However, comprehensive toxicity investigation of hydroxyapatite-based NPs is still needed.

## 5. Conclusions

Plant-derived natural products exhibit high potential as anticancer agents. The limited toxicity, affordability, and diversity in mechanisms of action are the main advantages of anticancer natural products. However, low bioavailability, low solubility, and limited stability reduced the use of these agents. Promising technologies were used in pharmaceutical modification of anticancer natural products, and the resulted formulations showed improved anticancer activities. However, many natural products extracted from plants exhibited high activity in vitro and in vivo. Unfortunately, there is a lack of clinical studies for some promising anticancer natural products. The production of therapeutic modalities based on natural products and synthetic analogs should be further expanded, to provide more therapeutic options for cancer.

## Figures and Tables

**Figure 1 molecules-25-05319-f001:**
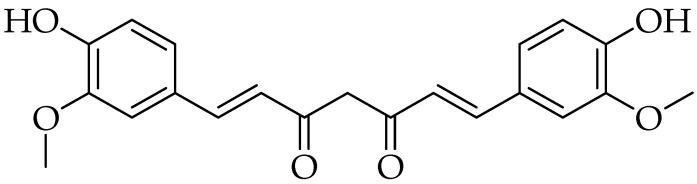
The chemical structure of curcumin.

**Figure 2 molecules-25-05319-f002:**
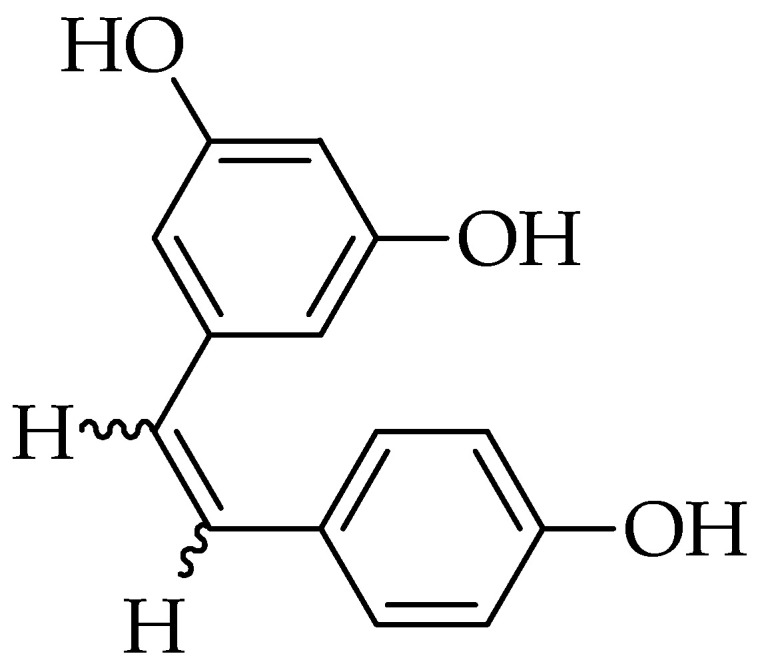
The chemical structure of resveratrol.

**Figure 3 molecules-25-05319-f003:**
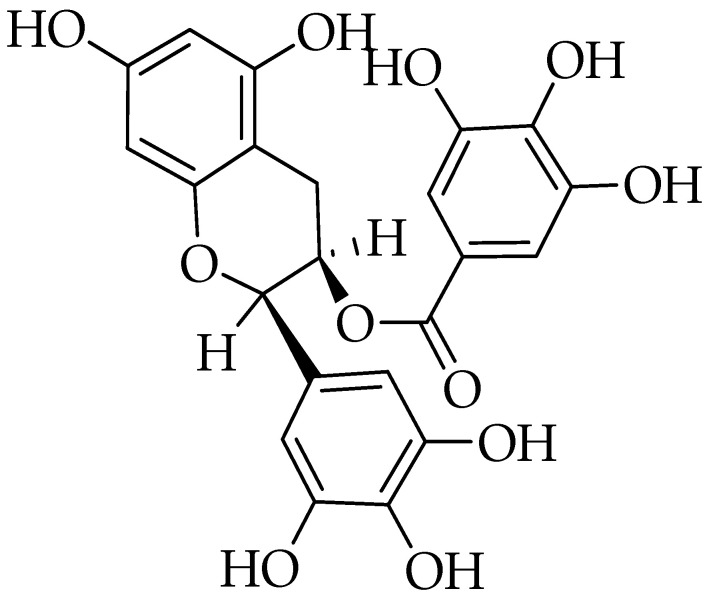
The chemical structure of epigallocatechin gallate.

**Figure 4 molecules-25-05319-f004:**
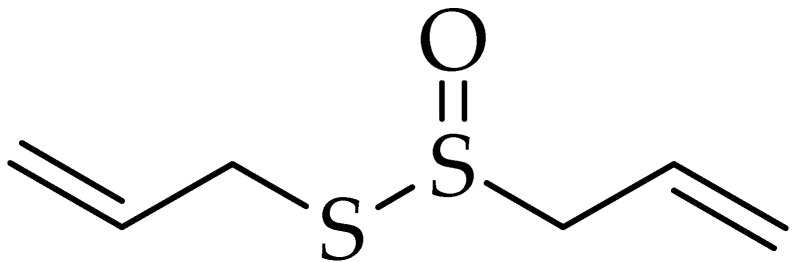
The chemical structure of allicin.

**Figure 5 molecules-25-05319-f005:**
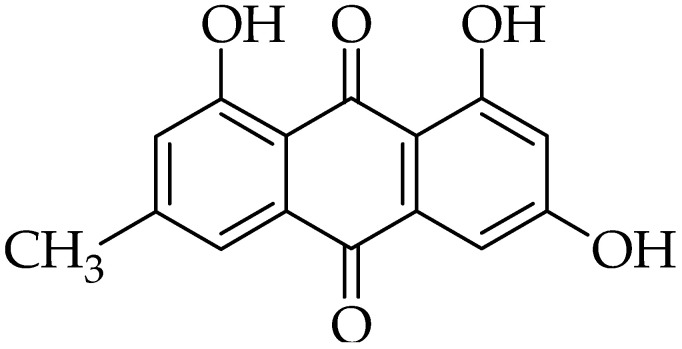
The chemical structure of emodin.

**Figure 6 molecules-25-05319-f006:**
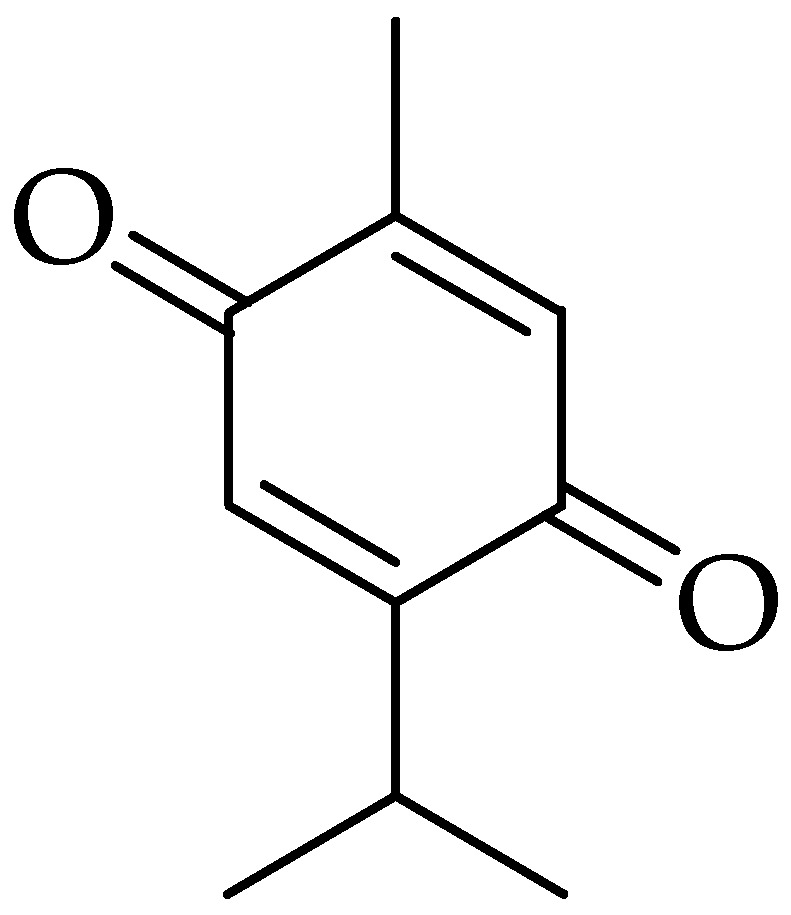
The chemical structure of thymquinone.

**Figure 7 molecules-25-05319-f007:**
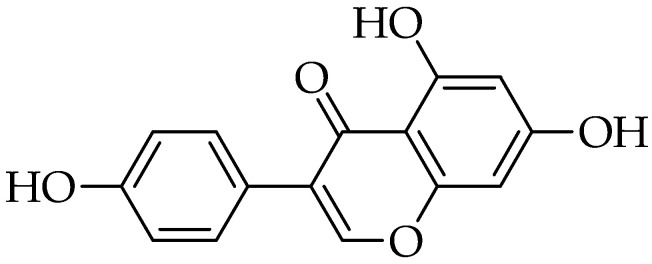
The chemical structure of genistein.

**Figure 8 molecules-25-05319-f008:**
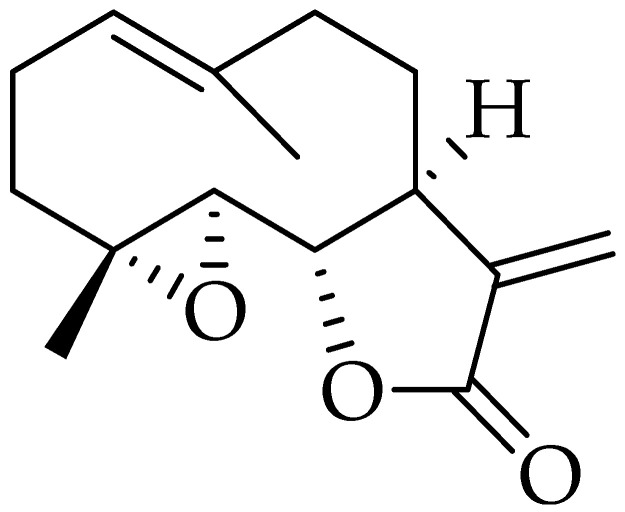
The chemical structure of parthenolide.

**Figure 9 molecules-25-05319-f009:**
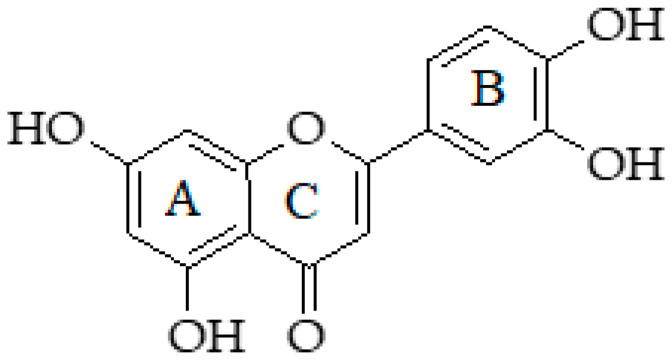
The chemical structure of luteolin.

**Figure 10 molecules-25-05319-f010:**
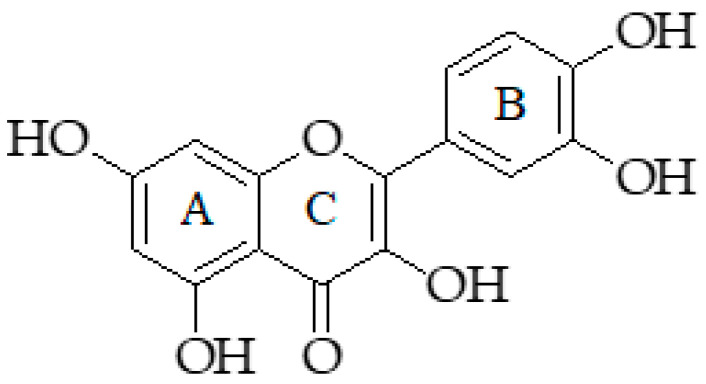
The chemical structure of quercetin.

**Figure 11 molecules-25-05319-f011:**
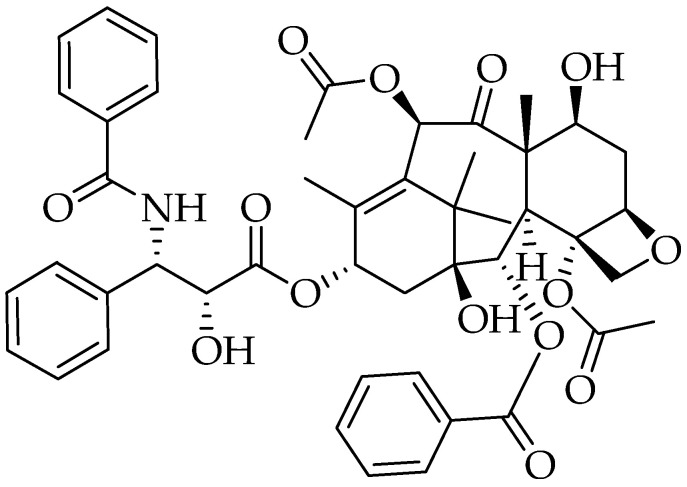
The chemical structure of paclitaxel.

**Figure 12 molecules-25-05319-f012:**
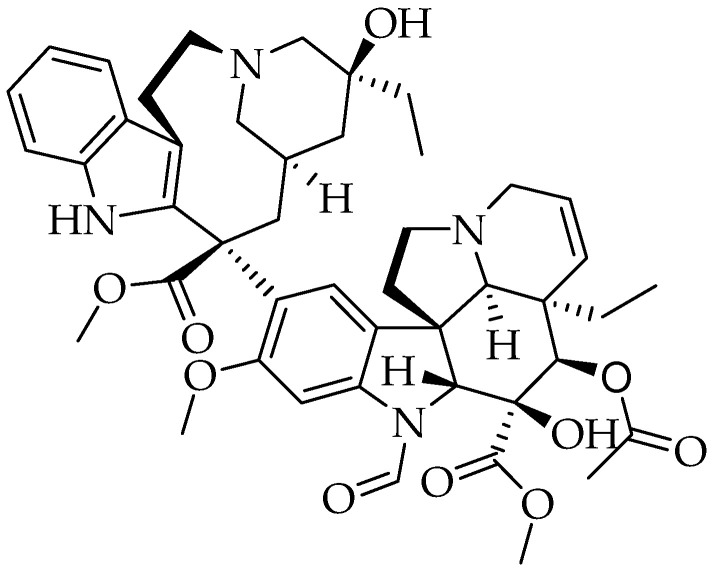
The chemical structure of vincristine.

**Figure 13 molecules-25-05319-f013:**
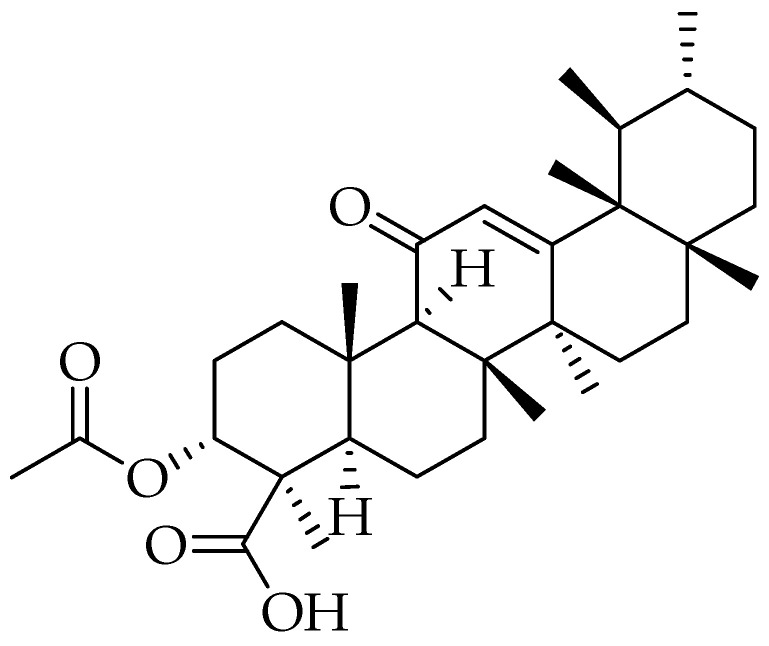
The chemical structure of acetyl-11-keto-β-boswellic acid.

**Table 1 molecules-25-05319-t001:** Mechanisms of action of natural products and their analogues.

Compounds	Cancer Cell Line and Animal Model	Mechanisms of Action	Classes of Analogues	Mechanisms of Action
Curcumin	HCT-116, HT-29, COLO-205, MG-63, U-87, U-251	Induced ligand (TRAIL) apoptotic pathways via upregulating death receptor 5 [31]. Initiated Fas-mediated apoptotic pathway by activating caspase-8 [32]. Upregulate Bax expression and suppress Bcl-2 through activation of p53 [33]. In activation of JAK/STAT signaling [14]. Inhibition of MMP-9; downregulating endothelial cell marker; and inhibition of STAT3 and NF-κB and activated caspase-3 [19].	4-bromo 4′-chloro analog 5,7-dimethoxy-3-(3-(2-((1*E*,4*E*)-3-oxo-5-(pyridin-2-yl)penta-1,4-dien-1-yl)phenoxy)propoxy)-2-(3,4,5-trimethoxyphenyl)-4*H*-chromen-4-one	Showed five-fold improvement in the potency and enhanced apoptosis via caspase-3 induction 19.9%, compared to the curcumin [393]. Enhanced cancer cell apoptosis through disruption of mitochondria function, prevented TrxR activity, and increased Bax/Bcl-2 production [394].
Resveratrol	HeLa DLD1, HCT-15 DBTRG HSC-3, HN-8, HN-30 PANC-1 MCF-7 T47D MDA-MB-321 Human melanoma cells A549 TC-1 mouse model BALB/c mice EAC mouse model	Inhibited cell growth by activating caspase-3 and caspase-9, upregulating of Bcl-2 associated X protein, and inducing expression of p53 [52]. Induced cell apoptosis and G_1_ phase arrest via suppression of AKT/STAT3 signaling pathway [53]. Improved apoptotic and oxidant effects of paclitaxel by activating TRPM2 channel [54]. Reduced vascular endothelial growth factor (VEGF) expression [55]. Inhibited metastasis by affecting IL-1β, TNF-α, vimentin, *N*-cadherin, and CTA-2 expressions [56]. Upregulating p53 and Bax expression, increasing Bcl-2 activity, and inducing caspase-3 activation. Decreased tumor size by downregulating E6 and tumor protein levels [57],	Imino-*N*-aryl-substituted 3,4′,5-*trans*-trimethoxystilbene ((*E*)-4,40-(ethene-1,2-diyl)bis(3-methylphenol))	Induced apoptosis by inhibition of topoisomerase II [395]. Improved anticancer properties of the natural resveratrol by inhibiting cell growth, preventing metastasis, and triggering cancer cells apoptosis [396]. Induced cell cycle arrest in S phase via modulation of cyclin A1/A2 and promoted cell death through upregulation of Bax/Bcl_2_ [397].
EGCG	NPC cells, NBC-39, HONE-1, NPC-BM HeLa PCa CAL 27 H1299 HSC-2 LNCaP HCT 15, HCT 116, Hep G-2 A549, H1299 BALB/c nu/nu mice EAC mouse model	Inhibited the metastatic activity by downregulation of protein expression of MMP-2 through modulation of the Src signaling pathway [82]. Downregulated cyclinD1 and upregulated cell cycle inhibitors LIMD1, RBSP3, and p16 at G1/S phase of the cell cycle [83]. Enhanced apoptosis by activating AKT/STAT3 pathway and suppressing multidrug resistance 1 signaling [85]. Inhibited cell growth through matrix metalloproteinase (MMP)-2- and -9-independent mechanisms [89]. Suppressed tumor growth in TRAMP mice and decreased tumor-derived serum PSA [91]. Inhibited cancer tumors in PDX model by suppressing the expression of NF-κB regulated genes [93].	DiestersG28, G37, and G56 Monoesters M1 and M2 Pro-EGCG EGCG-C16	Improved cancer cell death by inducing apoptosis and inhibition of FASN activity [398]. Inhibits cell proliferation via downregulation of cellular proteasome [399]. Prevents tumor growth by inhibiting the phosphorylation of EGFR, as well as inducing apoptosis [400].
Allicin	CCA HCT-116 U87MG U251-MG, A172 A375 DBTRG-05MG HGC27, AGS SK-MES-1, DLD-1 SK-Hep-1, BEL-7402 U251 MCF-7 EMT6/P Nude mouse model BALB/c mice	Induced apoptosis and prevented cell migration through upregulating of SHP-1 and inhibiting STAT3 activation [107]. Attenuated tumor growth in the nude mouse model of cholangiocarcinoma [107]. Inhibition of NF-κB signaling pathway [108]. Upregulates miR-486-3p and increases chemosensitivity to temozolomide [110]. Inhibition of cytokine release and upregulation of p53 activity [109]. Inhibiting ornithine decarboxylase, a rate-limiting enzyme in cell proliferation of neuroblastoma, and inducing cell apoptosis [111]. Suppresses melanoma cell growth via increasing cyclin D1 and reducing MMP-9 mRNA expression [112]. Inhibiting human glioblastoma proliferation by stimulating S and G_2_/M phase cell cycle arrest, apoptosis, and autophagy [113]. Reducing growth and metastasis through upregulation of miR-383-5p and downregulation of ERBB4 [114].	3f, 3h, 3m, and 3u	Increased caspase-3 activity and modulated Bax/Bcl_2_ expression [401].
Emodin	SW1990 HT-29, HUVECs SGC996 DU-145	Cell cycle arrest, apoptosis, and the promotion of the expression of hypoxia-inducible factor 1α, glutathione *S*-transferase *P*,*N*-acetyltransferase, and glutathione phase I and II detoxification enzymes, while inhibiting angiogenesis, invasion, migration, chemical-induced carcinogen-DNA adduct formation, HER2/neu, CKII kinase, and p34cdc2 kinase [136]. Inhibits tumor-associated angiogenesis through the inhibition of ERK phosphorylation [137]. Downregulates the expression of survivin and β-catenin, inducing DNA damage and inhibiting the expression of DNA repair [136,138]. Inhibits the activity of casein kinase II (CKII) by competing at ATP-binding sites [136,139]. Upregulates hypoxia inducible factor HIF-1 and intracellular superoxide dismutases and boosts the efficacy of cytotoxic drugs [140,141]. Decreases the expression of MDR-1 (*P*-gp), NF-κB and Bcl-2 and increasing the expression levels of Bax, cytochrome-C, caspase-9 and -3, and promoting cell apoptosis [142]. Downregulates both XIAP and NF-κB and enhances apoptosis [143,144]. ROS-mediated suppression of multidrug resistance and hypoxia inducible factor-1 in overactivated HIF-1 cells [146].	Em08red (1,8-dihydroxy-9(10*H*)-anthracenone)	Suppressed ErbB2 activity, triggered G2 arrest, downregulated the expression of (Bcl-xl and Bcl-2), and induced caspase-3 and caspase-9 [402].
Thymoquinone	HCT116, HCT116 P53, HepG2 MCF-7 PDA HGC27, BGC823, SGC7901 Wistar rats Wistar albino rats BALB/c mice	Upregulation of p21cip1/waf1 and a downregulation of cyclin E, and associated with an S/G2 arrest of the cell cycle [154]. Induced the G0/G1 cell cycle arrest, increased the expression of p16, decreased the expression of cyclin D1 protein, inactivated CHEK1, and contributed to apoptosis [155,156]. Reduced the elevated levels of serum TNF-α, IL-6, and iNOS enzyme production [158]. Reducing the NO levels by downregulation of the expression of iNos, reducing Cox-2 expression, and consequently generating PGE2 and reducing PDA cells synthesis of Cox-2 and MCP1 [159,160]. Noticeably reduced the phosphorylation of EGFR at tyrosine -1173 residues and JAK2 [161]. Elevation of PPAR-γ activity and downregulation of the gene’s expression for Bcl-2, Bcl-xL, and surviving [162]. Downregulation of the expression of STAT3-regulated gene [163]. Activation of caspases 8, 9, and 7 in a dose-dependent manner and increases the activity of PPAR-γ [165,167]. Decrease of expressions of CYP3A2 and CYP2C 11 enzymes [171]. Increase of the PTEN mRNA [173]. Suppresses androgen receptor expression and E2F-1 [148].	Analogues 6 and 14 ATQTHB and ATQTFB	Inhibited cancer cell growth two-fold, compared to the natural thymoquinone [403]. Suppresses cell viability and reduces the pro-survival and pro-angiogenic molecules COX-2 [404].
Genistein	LNCaP, PC3, DU-145 PNT-2, VeCaP MDA-MB-231 T47D HT29, COLO201 A549, NCI-H460 BxPC-3, PANC-1 MML-1, SK-MEL-2 U87, LN229 HeLa, CaSki MCF-7 Nude mice model Wistar rats Lobund-wistar rat	Inhibits cyclooxygenase-2 (COX-2) directly and indirectly by suppressing COX-2-stimulating factors like activated protein-1 (AP-1) and Nf-κb [175]. Inhibits CDK by upregulating p21; suppresses cyclin D1, ultimately inducing G2/M cell cycle arrest; and decreases tumor cell progressions [175,184,186,187,188]. Downregulates the expression levels of matrix metalloproteinase-2 (MMP-2) [175,189,190]. Inhibits several targets, including Cyclin D1, MMP, VEGF, Bcl-2, uPA, and Bcl-XL [175]. Influences metastasis and induces apoptosis by inhibiting Akt, as well as NF-κB cascades [175,191]. Inhibits histone deacetylase (HDAC) enzymes, which are responsible of regulating histone acetylation of DNA [175]. Inhibition of Hsp90 chaperones [197].	DFOG (7-difluoromethoxyl-5,4′-di-*n*-octylgenistein)	Reduces expression of c-Myc and P13k/AKT [405].
Parthenolide	SCK, JCK, Cho-CK, Choi-CK BT20 MDA-MB-231 MDA-MB436 U87MG, U373 TRAMP mice C57BL/6 mice Kras^G12D/+^; LSL-Trp53^R172H^; Pdx-1-Cre mouse model	It mediated STAT3 inhibition, inducing the expression of death receptors and, hence, an apoptotic pathway [216]. Activation of p53 and the increased production of reactive oxygen species (ROS) [199,210], along with reduced glutathione (GSH) depletion [214]. Targets mitochondrial thioredoxin reductase to elicit ROS-mediated apoptosis [217]. Interferes with microtubule formation and prevents proliferation of malignant cells [218]. Induces thrombopoiesis through the inhibitory activity of NF-κB and consequently renders cancer cells prone to undergo apoptosis [219]. Impairs focal adhesion kinase-dependent signaling pathways and, hence, the cell proliferation, survival, and motility [220].	(−)-goyazensolide, (−)-15-deoxygoyazensolide DMAPT	Reduced cancer cell viability through activation of caspase-3 and suppression of NF-κB [406]. Induces apoptosis via stimulation of ROS and inhibition of NF-κB [407].
Luteolin	MDA-MB-231 BT5-49 LoVo A549, H460 SCC-25 Xenograft metastasis mouse model Swiss albino mice	Activates both the extrinsic and intrinsic apoptosis pathways and increases the expression of death receptor 5 [243]. Inhibits cell growth and induces G2 arrest and apoptotic cell death via activating JNK and inhibiting translocation of NF-κB [244]. Suppressed proliferation and survival of cancer cells by inhibition of angiogenesis through blocking activation of the VEGF receptor and its downstream molecule PI3K/Akt and PI3K/p70S6 kinase pathways [245]. Inhibitions of wide panel of receptor tyrosine-kinases activity, such as human epidermal growth-factor receptor 2 (HER-2), insulin-like growth factor (IGF), and epidermal growth-factor receptor (EGFR) [246]. Increased the expression of genes related to apoptosis and stress response within LC540 tumor Leydig cells [249].	NA	
Quercetin	MCF-7, MDA-MB-231 CT26, MC38, CACO OV2008, SKOV3 A549 T24 UMUC3, MB49 MC3T3-E1 Swiss albino mice	Reduced the expression of epidermal growth factor receptor (EGFR), tyrosine kinases involved in the development of a wide variety of solid tumors, resulting in the inhibition of cell growth and the induction of apoptosis [272]. Increased the expression of death receptor 5 (DR5) resulting in stimulation of tumor necrosis factor related apoptosis-inducing ligand (TRAIL) and subsequent cancerous cells apoptosis [273]. Direct targeting of Raf and MEK in Raf/MEK/ERK cascade, which is important pathway in neoplastic transformation [274]. Activation of caspases cascade; increases the level of caspase-3 and -9 and then higher expression of proapoptotic Bcl-2 family members and lower levels of antiapoptotic Bcl-xL that contributed directly to the apoptotic process [275]. Interaction of quercetin with DNA directly as one of the mechanisms for inducing [276].	Q3’S, Q3G Q2, and Q5	Stimulates cell cycle arrest in S phase and activates ROS-dependant apoptosis pathway [408]. Triggered apoptosis via suppression of topoisomerases and activation of ROS pathway [271].
Paclitaxel	HeLa Mouse fibroblast cell	Stabilization of cellular microtubules through binding β-tubulin subunit and inhibiting their depolymerization leading to block in the progress of mitotic division and prohibit cell division to ultimately cause apoptosis [307]. Causes cell death due to chromosome miss-aggregation on multipolar spindles where the resultant daughter cells are aneuploid, and a portion of these die due to loss of one or more essential chromosomes [308]. Targets the mitochondria and inhibits the function of the apoptosis inhibitor protein B-cell Leukemia 2 (Bcl-2) [309].	Docetaxel	Disruption of microtubular depolymerization and modulation of bcl-2 and bcl-xL gene expression [409].
Vincristine	mice transplantable leukemia P-1534	Inhibition of polymerization of the microtubules through binding with the tubulin. This produces an arrest in G2/M phase and induces apoptosis [325]. Inhibitor of topoisomerase II [326]. High affinity to chromatin; binding of vincristine alters chromatin structure that perturbs histone-DNA interaction and possibly removal/displacement of the histones from DNA is occurred resulting in increasing of its cytotoxic effect [327].	Vinblastine	Inhibition of cell division via interaction with tubulin formation, resulting in mitotic arrest or cell death [410].
Bromelain	A431, A375 MCF-7 4T1 Swiss albino mice	Increases the expression of p53 and Bax activators genes of apoptosis in cancerous cells, and promotes apoptotic cell death in tumors [350]. Induced apoptosis via activating both caspase dependent and independent pathways [351]. Diminished the expression of the cell cycle regulatory proteins cyclin A, cyclin B, and cyclin D, resulting in G1 arrest [352]. Inhibition extracellular signal regulated protein kinase (ERK1/2) and p38 mitogen-activated protein kinase (MAPK), besides the decrease in Cox-2 expression and inhibition of NF-κB pathway [353]. Antiangiogenic effect by interfering with VEGF [354,355]. Stimulates ROS, and this would have a direct impact on the modulation of signaling in cancer cells, leading to tumor-cell-killing properties [334]. Upregulation of c-Jun *N*-terminal kinase and p38 kinase [357].	NA	
Boswellic acid	PC-3 HT-29, HCT-116, LS174T HL-60 A549, H460, H1299 HCT-8/VCR HCC	Inhibition of topoisomerases I and II, leading to apoptosis in different cell lines [376,377]. Downregulation of G1 phase cyclins and cyclin-dependent kinases (CDK) [378]. Induced apoptosis accompanied by activation of caspase-3, -8, and -9, resulting in expression of DR4 and DR5 [379,380,381]. Suppressed tumor growth through inhibition of angiogenesis by targeting vascular endothelial growth factor (VEGFR2) signaling pathway [382]. Prohibited the phosphorylation of extracellular-signal-regulated kinase-1 and -2 (Erk-1/2) and impaired the motility of cancer cells; Erk pathway plays a crucial role in signal transduction and tumorigenesis [383]. Inhibiting autophagy through regulating the ERK and P53 signaling pathways [384]. Suppressed TNF-induced invasion through inhibition of NF-κB regulated gene expression [385]. DNA damage response accompanied by impairment of DNA repair genes [386].	Analogues 7, 8, 9, and 10	Induce cancer cell death by promoting DNA fragmentation [410].

**Table 2 molecules-25-05319-t002:** Commercially approved target delivery systems and their active componenets.

Trade Name	Active Substance	Drug Delivery System	Indications
Lipusu^®^	paclitaxel	Lecithin/cholesterol liposom	Treatment of ovarian, breast, non-SCLC, gastric, and head and neck cancer [656]
Abraxane^®^	paclitaxel	Nanoparticle albumin-bound	Metastatic adenocarcinoma of the pancrease and breast cancer [658]
Opaxio^®^	paclitaxel	Polymer-based nanoformulation	Treatment of glioblastoma [699]
Marqibo^®^	Vincristine	sphingomyelin/cholesterol (SM/Chol) liposom	Treatment of adults with relapsed ALL [665]
Theracurmin^®^	Curcumin	Colloidal dispersion using ghatti gum and glycerin	Improve health life quality and work as antioxidant, as well as anti-inflammatory [700]
Meriva^®^	Curcumin	Curcuminoids and phosphatidylcholine phytosome	Improve health life quality, anti-inflammatory effect in patients with solid tumors [701]

ALL = acute lymphoblastic leukemia.

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
