# Peer review of "Plant-Derived Natural Products in Cancer Research: Extraction, Mechanism of Action, and Drug Formulation"

_molecules, 2020, doi:10.3390/molecules25225319_

Round 1

Reviewer 1 Report

This revision authored by Talib et al is interesting, well written and easy to follow. However, I recommend authors include more figures showing more chemical structures in each sections as well as schemes when they refer to drug delivery systems such as liposomes, or other systems.

Also, it seems authors focused only in the most common natural products with anticancer activities. There are lots of other examples, including analogues of the ones that are described in their manuscript.

I also recommend the inclusion of tables for classes of analogues and their mechanisms of action.

Authors should standardize all chemical structures, they are in different sizes and styles.

English should be revised,

Author Response

Thank you for your constructive comments. All your comments were considered and changes are highlighted in yellow color.

Reviewer 2 Report

The manuscript deals with the extraction, modes of action and drug formulation of 14 anti-cancer active compounds of natural origin.

Overall, the manuscript is interesting and original. It is nicely written and seems suitable for publication in Molecules. There are however few minor issues which should be addressed more carefully prior the final acceptance:

  • The Figure 1 should be changed/rearranged, or split into a set of separate figures. Providing the structures of compounds discussed is highly desirable, but the figure should definitely be placed earlier in the text. To be honest, a separate figure depicting each individual compound implemented in the relevant subsections of the main Section 2 would look much better. This simple operation would result in the much more transparent and informative way for discussing the compounds mentioned.
  • An implementation of a general table consisting of the list of compounds, cancer cell lines tested, and activity values would be more than welcome.

Accordingly, I recommend a minor revision of the manuscript.

Author Response

Thank you for your constructive comments. All your comments were considered and changes were highlighted in yellow color inside the manuscript.

Reviewer 3 Report

Comments for molecules-978445

The manuscript reviewed the main plant-derived natural products used as anticancer agents. Natural sources, extraction methods, anticancer mechanisms, clinical studies, and pharmaceutical formulation were discussed in this review. The content of the paper is quite valuable, but there are some problem in experimental design should be solved before the manuscript been considered for publication.

Substantial revisions

Q1: Please add the name of (H)~(M) chemical formula in Figure 1.

Q2: Please add table for the commercially approved target delivery and natural substances, for readers to read and understand.

Reference: (Example)

Nadine Wiesmann, Wolfgang Tremel b and Juergen Brieger. Zinc oxide nanoparticles for therapeutic purposes in cancer medicine. J. Mater. Chem. B, 2020, 8, 4973—4989.

Q3: Although 3.1.1 nanoparticles (NPs) have been partially approved for sale, related studies nanoparticles size, concentration, time course and the stabilizer used still damage cells or tissues (Bahadar.H et al. al., 2016; Malhotra,y N., et al., 2020 ). In addition, Liu et al. observed that magnetic iron oxide nanoparticles accumulate and inhale in the liver, spleen, lung and brain, indicating that they have the ability to cross the blood brain Barrier ability (Liu G et al., Small, 2013). Please add the literature discussion on the toxicity and safety considerations of Nanoparticles (NPs).

(Bahadar. H et al., 2016)

(Malhotra,y N., et al., 2020)

Reference:

Nemi Malhotra,y, Jiann-Shing Lee ,y, Rhenz Alfred D. Liman , Johnsy Margotte S. Ruallo , Oliver B. Villaflores , Tzong-Rong Ger , and Chung-Der Hsiao. Potential Toxicity of Iron Oxide Magnetic Nanoparticles: A Review. Molecules 2020, 25, 3159; doi:10.3390/molecules25143159.

Haji Bahadar, Faheem Maqbool, Kamal Niaz1 and Mohammad Abdollahi. Toxicity of Nanoparticles and an Overview of Current Experimental Models. Review article Iranian Biomedical Journal 20(1): 1-11 January 2016.

Liu G, Gao J, Ai H, Chen X. Applications and potential toxicity of magnetic iron oxide nanoparticles. Small, 2013; 9(9-10): 1533-1545.

Author Response

Thank you for your constructive comments. All comments were considered and changes were highlighted in yellow color inside the manuscript.

Round 2

Reviewer 1 Report

Authors had substantially modified this revision and now, in this new version, it is satisfactory for publication in Molecules.

Reviewer 3 Report

Comments for molecules-978445

The manuscript reviewed the main plant-derived natural products used as anticancer agents. Natural sources, extraction methods, anticancer mechanisms, clinical studies, and pharmaceutical formulation were discussed in this review. The content of the paper is quite valuable. In the revised manuscript, My questions in experimental design has been responded by authors. Thus I suggest that the manuscript be considered for publication.